# The ELAVL3/MYCN positive feedback loop provides a therapeutic target for neuroendocrine prostate cancer

Yiyi Ji[1,5], Weiwei Zhang [1,5], Kai Shen[1,5], Ruopeng Su[1], Xinyu Liu[1], Zehua Ma[1], Bo Liu[1], Cong Hu[1], Yizheng Xue[1], Zhixiang Xin[1], Yi Yang[1], Ang Li[1], Zhou Jiang[2], Na Jing[3], Helen He Zhu [3], Liang Dong[1], Yinjie Zhu[1], Baijun Dong[1], Jiahua Pan[1], Qi Wang [1,4] ✉ & Wei Xue[1] ✉

Neuroendocrine prostate cancer is a rapidly progressive and lethal disease characterized by early visceral metastasis, poor prognosis, and limited treatment options. Uncovering the oncogenic mechanisms could lead to the discovery of potential therapeutic avenues. Here, we demonstrate that the RNA-binding protein ELAVL3 is specifically upregulated in neuroendocrine prostate cancer and that overexpression of ELAVL3 alone is sufficient to induce the neuroendocrine phenotype in prostate adenocarcinoma. Mechanistically, ELAVL3 is transcriptionally regulated by MYCN and subsequently binds to and stabilizes *MYCN* and *RICTOR* mRNA. Moreover, ELAVL3 is shown to be released in extracellular vesicles and induce neuroendocrine differentiation of adenocarcinoma cells via an intercellular mechanism. Pharmacological inhibition of ELAVL3 with pyrvinium pamoate, an FDA-approved drug, effectively suppresses tumor growth, reduces metastatic risk, and improves survival in neuroendocrine prostate cancer mouse models. Our results identify ELAVL3 as a critical regulator of neuroendocrine differentiation in prostate cancer and propose a drug repurposing strategy for targeted therapies.

Neuroendocrine prostate cancer (NEPC) is an aggressive, early meta-static subtype that may occur de novo or arise in the late stages of disease progression after developing resistance to androgen deprivation therapy[1]. Although the application of next-generation androgen receptor inhibitors such as enzalutamide and abiraterone has successfully improved the survival of patients with advanced prostate cancer, these treatments have also been associated with an elevated incidence of NEPC[2,3]. NEPC distinguishes itself from conventional prostate adenocarcinoma through its histological attributes resembling small cell carcinoma and its expression of neuroendocrine markers including *CHGA*, *NSE*, *SYP*, and *NCAM1*[4–6]. Although the platinum-

based chemotherapy combined with etoposide currently stands as the recommended first-line treatment, this combination results in a median survival of only 10.5 months, which underscores the necessity for a deeper comprehension of the underlying disease mechanisms and the development of more efficient therapeutic strategies[6,7].

Comprehensive genomic analyses have revealed that NEPC is associated with frequent genomic abnormalities involving *PTEN* and *TP53*[8–10], which are commonly observed in most prostate adenocarcinomas, along with a specific prevalence of *RB1* loss in NEPC[4,11]. The combined deletion of these three tumor suppressors in mouse prostate epithelia resulted in the spontaneous initiation of prostate cancer

[1]Department of Urology, Ren Ji Hospital, Shanghai Jiao Tong University School of Medicine, Shanghai 200120, China. [2]Department of Pathology, Ren Ji Hospital, Shanghai Jiao Tong University School of Medicine, Shanghai 200120, China. [3]State Key Laboratory of Oncogenes and Related Genes, Ren Ji Med-X Stem Cell Research Center, Shanghai Cancer Institute & Department of Urology, Ren Ji Hospital, Shanghai Jiao Tong University School of Medicine, Shanghai, China. [4]Shanghai Key Laboratory for Tumor Microenvironment and Inflammation, School of Medicine, Shanghai Jiao Tong University, Shanghai 200120, China. [5]These authors contributed equally: Yiyi Ji, Weiwei Zhang, Kai Shen. ✉e-mail: wqi@sjtu.edu.cn; xuewei@renji.com

with neuroendocrine features, such as elevated expression of neuroendocrine markers in tumor tissues, increased risk of visceral metastasis, and resistance to castration treatment. Therefore, this mouse model serves as an ideal preclinical platform for research and facilitates the evaluation of therapeutic strategies for NEPC.

Previous studies have reported that *MYCN*, an oncogenic regulator involved in multiple neuroendocrine tumors, undergoes amplification and is overexpressed in 40% of NEPC cases, compared with only 5% incidence in prostate adenocarcinoma cases. As a transcription factor, MYCN preferentially binds to conserved promoter sequences, such as CACGTG, thereby instigating the activation of downstream genes. Genome-wide analysis and molecular characterization have identified many MYCN-target genes, encompassing oncogenic and neuroendocrine factors, such as *NSE* and *SYP*[4,12–15]. The overexpression of MYCN and activation of AKT1 can transform human prostate epithelial cells into tumors with neuroendocrine features, highlighting the crucial role of MYCN in the initiation of NEPC[16]. Despite its significance in NEPC, MYCN remains a challenging target for drug development in this disease context. Alternatively, numerous studies were eyeing the interactions and regulatory mechanisms associated with MYCN. AURKA, another key regulator of neuroendocrine tumors, binds to and stabilizes MYCN, prevents MYCN protein degradation, and thereby promotes cell proliferation in NEPC and other neuroendocrine tumors[12,17–19]. Several AURKA inhibitors have demonstrated anti-proliferative efficacy in preclinical models of neuroendocrine tumors by disrupting the MYCN-AURKA complex. Among them, alisertib has emerged as a promising candidate and entered a phase II clinical trial[20]. Although this study failed to reach the primary end point, it opened an avenue for therapeutic strategies that aim to target the MYCN interaction and regulation.

In this work, in order to better understand the cellular architecture associated with neuroendocrine differentiation, we reanalyze a previous single-cell RNA sequencing study using needle biopsies obtained from six patients with castration-resistant prostate cancer. Of the six patients, two had pathologically proven neuroendocrine prostate cancer[21]. Through the bioinformatics analysis of our dataset and the incorporation of other publicly available prostate cancer datasets[4,21–23], we identify *ELAVL3*, a member of the Embryonic Lethal Abnormal Vision-like RNA-binding protein family, as a key regulator implicated in the initiation and maintenance of neuroendocrine differentiation. We reveal a positive feedback loop, wherein MYCN transcriptionally upregulates *ELAVL3*, and conversely, ELAVL3 binds to and stabilizes *MYCN* mRNA. Most importantly, our findings demonstrate that disrupting the interaction between ELAVL3 and *MYCN* with pyrvinium pamoate, an FDA-approved drug, may offer a drug repurposing strategy with therapeutic potential for patients with NEPC.

# Results

## ELAVL3 expression is associated with neuroendocrine differentiation of prostate cancer

To pinpoint the crucial genes involved in the neuroendocrine differentiation of prostate cancer, we analyzed the single-cell RNA sequencing data from samples obtained from six castration-resistant prostate cancer patients, including two with pathologically confirmed neuroendocrine prostate cancer (NEPC)[21]. The unsupervised clustering analysis grouped 14210 epithelial cells into 16 clusters, two of which exhibited a high-neuroendocrine profile (Supplementary Fig. S1a). There were 227 genes in these two clusters that were expressed at higher levels than the remaining 14 clusters, which had lower neuroendocrine features. We then analyzed the 227 genes by integrating them with data from three additional publicly available transcriptome datasets related to prostate cancer obtained from both human and murine sources[4,21–23]. This integrated analysis identified nine genes that were consistently upregulated across all four datasets, as expected, including several well-known neuroendocrine-associated genes such

as *CHGA* and *ASCL1* (Fig. 1a). From the list of selected essential genes, we narrowed our focus to *ELAVL3*, a member of the Embryonic Lethal Abnormal Vision-like RNA-binding protein family (Fig. 1a). Upon further re-analysis of the above transcriptome data, we revealed that *ELAVL3* showed a positive correlation with several neuroendocrine biomarkers (*SYP*, *CHGA*, and *CHGB*), as well as transcription factors associated with neuroendocrine traits (*MYCN*, *NCAM1*, and *EZH2*). In contrast, it exhibited an inverse correlation with genes linked to androgen receptor, such as *AR*, *KLK3*, and *NKX3-1* (Fig. 1b and Supplementary Fig. S1a–c). Pathway enrichment analyses of the Beltran 2016[4] and TCGA[10] datasets implied that high expression of *ELAVL3* was associated with pathways related to neurons and oncogenesis, especially the PI3K/AKT/mTOR signaling (Supplementary Fig. S1d, e).

To assess the protein expression of ELAVL3 in neuroendocrine prostate cancer, we performed immunohistochemical staining using samples from our cohort of 196 patients with prostate cancer, including benign prostate tissues ($n = 15$), hormone-sensitive prostate cancer ($n = 144$), castration-resistant prostate adenocarcinoma ($n = 15$), and neuroendocrine prostate cancer ($n = 22$) (Fig. 1c; Supplementary Table 1). Quantitative analysis of the staining intensity and percentage revealed a discernible upregulation of ELAVL3 expression specific to neuroendocrine prostate cancer tissues (Fig. 1d). Further analysis of the transcriptome data from the Cancer Cell Line Encyclopedia also revealed a relatively higher basal level of *ELAVL3* in the NEPC-derived cell line NCI-H660 than in adenocarcinoma cell lines (Supplementary Fig. S1f). We then confirmed this result in a panel of prostate cancer cells and consistently observed high basal expression of ELAVL3 in NCI-H660 as well as in the neuroendocrine-associated cell line LASCPC-01, at both the RNA and protein levels (Fig. 1e, f).

In addition, we collected tumor tissues from two genetically engineered mouse models of prostate cancer, namely Pb-Cre4: *Pten*[f/f]; *Trp53*[f/f] (DKO) and Pb-Cre4: *Pten*[f/f]; *Trp53*[f/f]; *Rb1*[f/f] (TKO). Consistent with previous studies[8,22,24], DKO mice spontaneously developed adenocarcinoma prostate cancer characterized by androgen receptor-positive features, whereas TKO mice developed androgen receptor-negative neuroendocrine prostate cancer. We observed elevated Elavl3 expression in the tumor tissues of TKO mice, as demonstrated in both primary and metastatic sites, when compared with tumor tissues derived from DKO mice and normal prostate tissues from their non-Cre littermates (Fig. 1g, h and Supplementary Fig. S1g). These findings indicate a consistent correlation between increased ELAVL3 expression and neuroendocrine differentiation in prostate cancer.

## ELAVL3 is essential for the development and maintenance of neuroendocrine prostate cancer

To determine the function of ELAVL3 in neuroendocrine prostate cancer, we carried out ribonucleoprotein immunoprecipitation sequencing (RIP-seq) on PC3 and DU145 cells that overexpressed ELAVL3. Our analysis discovered 1142 transcripts that bound to ELAVL3 in at least one of the examined cell lines, mainly related to RNA-specific processes including mRNA 3′-UTR AU-rich region binding, and neuronal-associated processes including neuron projection development and synapse organization, as evidenced by Gene Ontology analysis (Supplementary Fig. S2a). To examine the biological significance of ELAVL3 in both cell lines, we compared the ribonucleoprotein immunoprecipitation (RIP) targets of the two cell lines and identified 310 overlapping genes (Fig. 2a). Notably, when subjected to hallmark gene sets and KEGG pathway analysis using Metascape[25], these shared genes demonstrated significant enrichment within oncogenic pathways, such as epithelial-mesenchymal transition, KRAS, IL6/JAK/STAT3, and the mTORC1 pathways (Fig. 2b), suggesting that ELAVL3 may function as a critical regulator in the progression of prostate cancer.

Therefore, we examined the impact of ELAVL3 overexpression in prostate cancer cell lines and discovered that it effectively stimulated

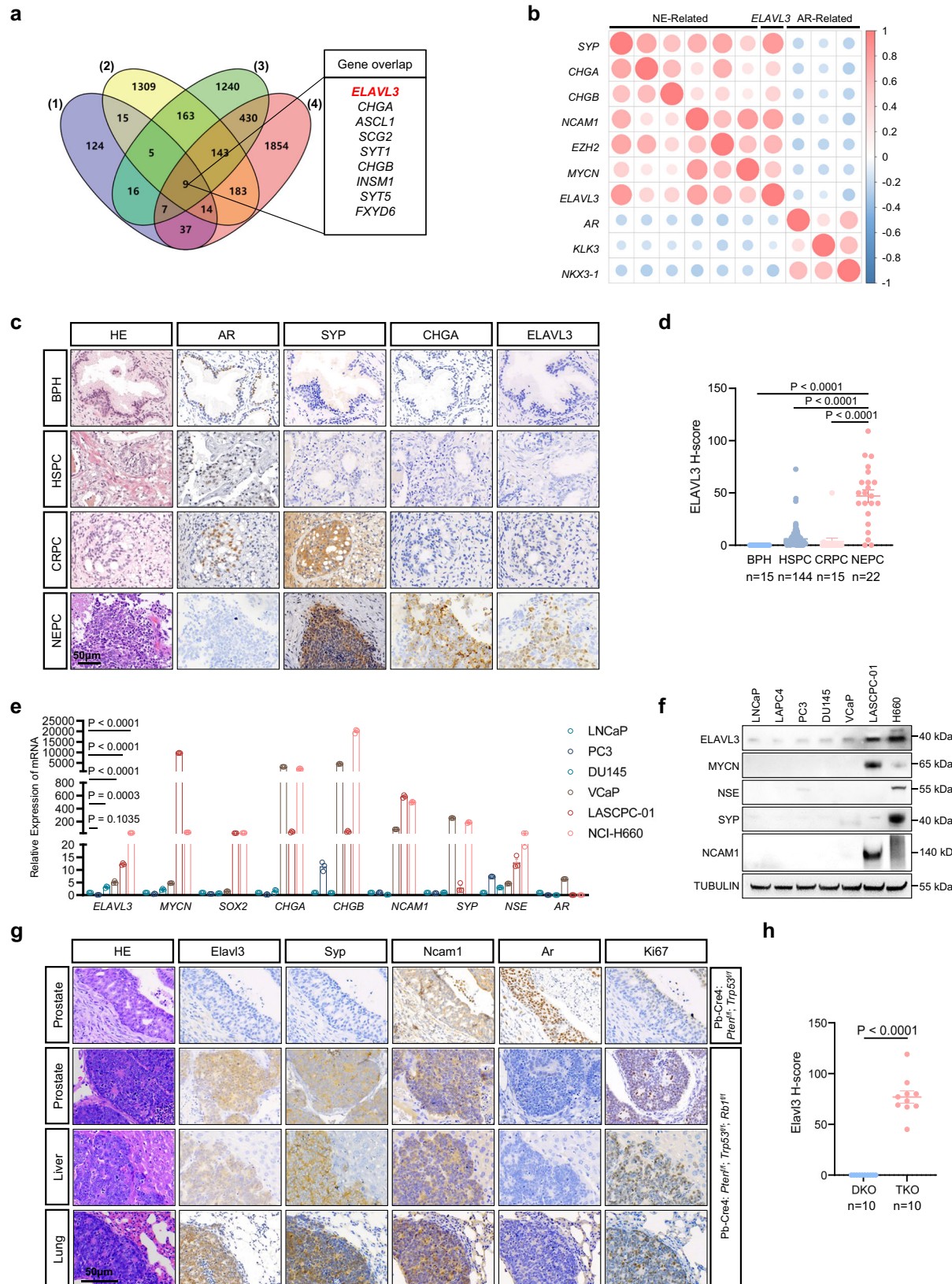

the expression of neuroendocrine markers, alongside MYCN, NCAM1, and SYP when compared with vector control (Fig. 2c and Supplementary Fig. S2b). Intriguingly, we also found that ELAVL3 overexpression activated the PI3K/AKT/mTOR pathway, as indicated by the phosphorylation of both AKT and S6, which have been previously identified as key pathways of neuroendocrine differentiation[4,16]. These

results further support the findings derived from our KEGG pathway analysis (Supplementary Fig. S1d, e) and suggest that ELAVL3 contributes to the activation of the PI3K/AKT/mTOR pathway, which will be discussed later. In contrast, the knockdown of ELAVL3 with two independent shRNAs reduced the expression of neuroendocrine-related genes in NCI-H660 and LASCPC-01 cells (Fig. 2d, e and

**Fig. 1 | ELAVL3 expression is associated with neuroendocrine differentiation of prostate cancer. a** Venn diagram showing the overlap of upregulated differentially expressed genes for the four NEPC RNA-seq datasets: (1) neuroendocrine prostate cancer (NEPC) versus castration-resistant prostate cancer (CRPC), data obtained from Beltran 2016 cohort[23]; (2) Pb-Cre4: *Pten*^f/f; LSL-*MYCN*^+/+ versus Pb-Cre4: *Pten*^f/f; LSL-*MYCN*^−/− organoids, data obtained from GSE86532[4]; (3) Pb-Cre4: *Pten*^f/f; *Rb1*^f/f versus Pb-Cre4: *Pten*^f/f, data obtained from GSE90891[22]; (4) two NE-high single-cell clusters versus 14 NE-low single-cell clusters, data obtained from GSE137829[21]. Common genes across all analytes are indicated. **b** Dot plots illustrating the Pearson correlation analysis between the expression of *ELAVL3*, NE-related genes, and AR-related genes. Data obtained from Beltran 2016 cohort[23]. **c** Representative H&E and immunofluorescence staining of AR, SYP, CHGA, and ELAVL3 in prostate tissues from indicated patient groups (BPH (n = 15), HSPC (n = 144), CRPC (n = 15), and NEPC (n = 22)). Scale bar, 50 μm. **d** Quantification of ELAVL3 staining in prostate tissues from indicated patient groups, each dot represents tissue from an individual patient. **e** QPCR showing relative mRNA expression of *ELAVL3*, NE-related genes

(*MYCN, SOX2, CHGA, CHGB, NCAM1, SYP*, and *NSE*), and *AR* in a panel of prostate cancer cells (n = 3 biologically independent experiments). **f** Western blot showing protein expression of ELAVL3 and NE-related genes (MYCN, NSE, SYP, and NCAM1) in a panel of prostate cancer cells. **g** Representative H&E and immunofluorescence staining of Elavl3, Syp, Ncam1, Ar, and Ki67 in tumors from mice of Pb-Cre4: *Pten*^f/f; *Trp53*^f/f (n = 10) and Pb-Cre4: *Pten*^f/f; *Trp53*^f/f; *Rb1*^f/f (n = 10). Scale bar, 50 μm. **h** Quantification of Elavl3 staining in prostate tumor tissues from mice groups: Pb-Cre4: *Pten*^f/f; *Trp53*^f/f (n = 10) and Pb-Cre4: *Pten*^f/f; *Trp53*^f/f; *Rb1*^f/f (n = 10); each dot represents tissue from an individual mouse. Data presented as mean ± s.e.m (**d, e, h**). Statistical significance was determined by two-tailed unpaired Student's *t*-test (**e, h**) or one-way analysis of variance (ANOVA) with Dunnett's multiple comparisons (**d, e**). Western blot experiments were repeated three times independently, with similar results (**f**). NE-related, related to neuroendocrine differentiation. AR-related, related to the androgen receptor pathway. BPH benign prostatic hyperplasia. HSPC hormone-sensitive prostate cancer. Source data are provided as a Source Data file.

Supplementary Fig. S2c). We generated neuroendocrine-like prostate cancer cells using previously established methods[4,26–28]. One method involved ectopic expression of MYCN in LNCaP or PC3 cells, while the other involved knockdown of TP53 and RB1 in LNCaP/AR cells, termed LNCaP/AR/shTP53/shRB1 cells (LAPR). As shown in Fig. 2f and Supplementary Fig. S2d–f, ELAVL3 deficiency reduced the upregulation of the neuroendocrine phenotype in these induced cell lines. Furthermore, our KEGG pathway enrichment analysis in LNCaP/AR/shTP53/shRB1 cells revealed a correlation between ELAVL3 deficiency and the reduction of several oncogenic pathways (MAPK, NFκB, Small cell lung cancer, and TNFα pathways) and neuronal-related pathways (GABAergic synapse, serotonergic synapse, glutamatergic synapse, and cholinergic synapse) (Fig. 2g). These findings suggest that ELAVL3 plays a crucial role in regulating neuroendocrine differentiation and oncogenic traits in prostate cancer.

Next, we examined the requirement of ELAVL3 for tumor cell proliferation in NEPC by investigating the effect of knocking down ELAVL3 on cell viability in LASCPC-01 cells. As shown in Supplementary Fig. S2g, the knockdown of ELAVL3 significantly reduced cell viability when compared with control shRNA. Next, we subcutaneously transplanted LASCPC-01 cells into castrated BALB/c nude mice (n = 6 per group) and found that ELAVL3 deficiency remarkably reduced both the weight and volume of the LASCPC-01 xenografts, as compared with the control group (Fig. 2h). Immunohistochemistry analysis of these tumor sections revealed that ELAVL3 deficiency inhibited cell proliferation, induced apoptosis, and caused a lineage switch from the neuroendocrine to luminal phenotype (Fig. 2i). The specificity of the ELAVL3 antibody used in this study was confirmed through immunohistochemical staining by blocking epitope with recombinant ELAVL3 protein (Supplementary Fig. S2h).

Given that neuroendocrine features are generally considered as one of the mechanisms contributing to androgen deprivation therapy resistance[4,29,30], we investigated whether ELAVL3 can modulate the therapy response in prostate cancer cells. To validate this hypothesis, we conducted RNA sequencing on LNCaP/AR/shTP53/shRB1 cells with shELAVL3 and shRNA control. Gene Set Enrichment Analysis revealed an enrichment of the androgen response signature in ELAVL3-deficient cells (Supplementary Fig. S2i, j). We found that the knockdown of ELAVL3 slightly inhibited cell viability (Fig. 2j), and this inhibitory effect was enhanced when cells were exposed to enzalutamide, indicating that ELAVL3 deficiency may sensitize cells to this therapy. Conversely, the overexpression of ELAVL3 in LNCaP cells conferred a heightened resistance to enzalutamide over time and showed a higher IC$_{50}$ (74.17 μM) than the control (23.55 μM) (Fig. 2k, l). These results suggest that ELAVL3 promotes the proliferation of neuroendocrine prostate cancer cells and, to a limited extent, decreases the cell's response to enzalutamide therapy.

## MYCN transcriptionally upregulates *ELAVL3* expression

The results of the above experiments showed a correlation between MYCN and ELAVL3: the ectopic expression of MYCN led to an increase in both the RNA and protein levels of ELAVL3, while the knockdown of ELAVL3 reversed the MYCN-mediated upregulation of neuroendocrine differentiation (Supplementary Fig. S2d, e). Due to its critical role as a transcription factor in neuroendocrine differentiation, MYCN directly binds to the promoter regions of several neuroendocrine markers, including *NSE* and *SYP*[4,12–15]. We reasoned that MYCN might also be implicated in the transcriptional upregulation of ELAVL3 in NEPC cells. To explore this hypothesis, we conducted promoter scanning in the JASPAR Database and identified three highly probable MYCN binding sites (BS) within the *ELAVL3* promoter region, denoted as BS1, BS2, and BS3 (Supplementary Table 2). We also analyzed an existing ChIP-seq dataset to examine chromatin occupancy of MYCN and revealed marked enrichment of MYCN binding sites in the promoter region of *ELAVL3*, but not *ELAVL4* (Fig. 3a)[28]. As shown in Fig. 3b, the presence of the MYCN-binding motif CACGTG in the *ELAVL3* promoter exhibited a high degree of conservation across species. To elucidate the molecular basis of MYCN transcriptional regulation, we cloned an *ELAVL3* promoter region encompassing approximately −2000 bp to +100 bp into a luciferase reporter, named the full-length construct. Then, three distinct promoter regions were identified and isolated from the full-length construct, and labeled as Segments A, B, and C. These segments were subsequently cloned into a luciferase reporter plasmid (Supplementary Fig. S3a). As shown in Fig. 3c, MYCN effectively activated the luciferase signal driven by both full-length *ELAVL3* and *ELAVL3* Segment A in HEK293T cells. Notably, Segment A includes two putative MYCN binding sites, BS1 and BS2, both of which contain the consensus sequence CACGTG. Furthermore, ChIP-qPCR analysis revealed that MYCN could directly bind to BS1 in LNCaP cells (Fig. 3d). Moreover, we generated three mutated luciferase constructs by replacing the consensus sequence CACGTG with CTGCGG at BS1, BS2, or both sites. A mutation in either BS1 or BS2, as demonstrated by signaling activity, was sufficient to disrupt the binding of MYCN. The combined mutation had a more prominent inhibitory effect on the binding when compared with the double-wildtype construct (Fig. 3e). To gain deeper insight into the influence of MYCN on *ELAVL3* transcription, we used the CRISPR/Cas9 technique to target the endogenous MYCN-binding site within the promoter region of *ELAVL3* (Supplementary Fig. S3b). Subsequently, we used 4-thiouridine (4sU) to label the nascent RNA, which was then isolated through biotin pull-down and quantified via qPCR. As shown in Fig. 3f, while ectopic expression of MYCN increased nascent *ELAVL3* transcription, CRISPR/Cas9-mediated deletion of MYCN-binding sites in the *ELAVL3* promoter effectively counteracted this elevation. These findings collectively indicate that *ELAVL3* is a direct transcriptional target of MYCN.

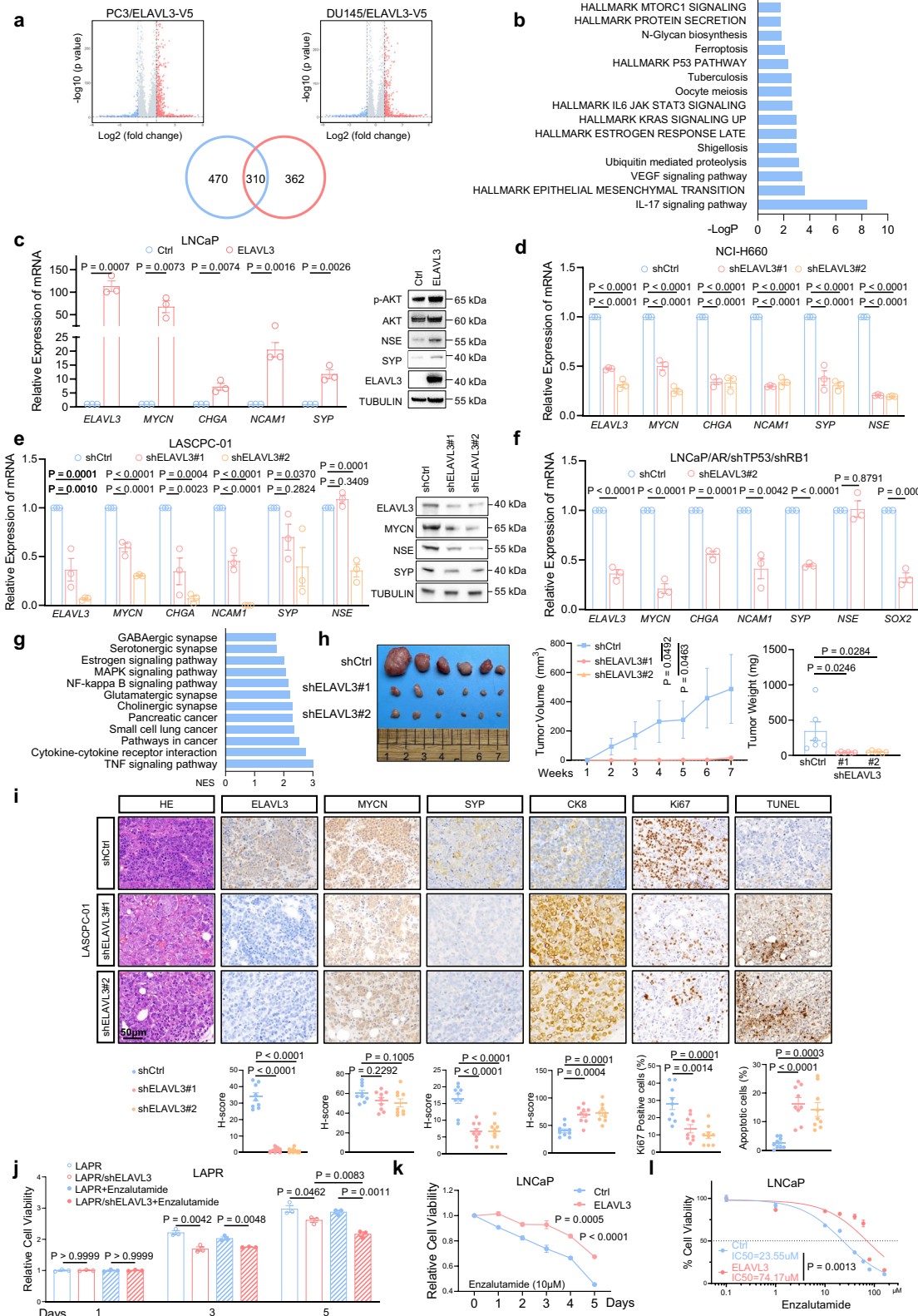

## ELAVL3 binds and stabilizes the mRNA of *MYCN*

Our observations extended to the modulation of MYCN expression: the ectopic expression of ELAVL3 increased both the RNA and protein levels of MYCN, while the knockdown of ELAVL3 resulted in a decrease in MYCN expression. Furthermore, MYCN deficiency repressed the neuroendocrine phenotype induced by ELAVL3, suggesting that MYCN

is required for ELAVL3 to function (Fig. 4a). Indeed, ELAVL3 over-expression sensitized LNCaP cells to MLN8237, an inhibitor targeting the MYCN/AURKA pathway, with an IC$_{50}$ of 0.067 μM compared with IC$_{50}$ = 1.084 μM for the control cells (Fig. 4b), indicating an increased reliance on the MYCN pathway in the presence of ELAVL3. Given that ELAVL3 is known to bind AU-rich elements in the 3′-UTR of

**Fig. 2 | ELAVL3 is essential for the development and maintenance of neuroendocrine prostate cancer. a** Volcano plots showing enrichment of RNA most significantly bound to ELAVL3-V5 in PC3 (left) and DU145 (right). Fold change >2; adjusted *P*-value < 0.05, calculated by wald test. Venn diagram showing overlap between putative ELAVL3 mRNA targets from PC3 and DU145. **b** Hallmark and KEGG pathway enrichment analysis of samples with transcripts simultaneously bound to ELAVL3-V5 from PC3/ELAVL3-V5 and DU145/ELAVL3-V5 using Metascape[25]. **c** QPCR showing relative mRNA expression of indicated genes (left) and Western blot showing indicated protein expression (right) in LNCaP/Ctrl and LNCaP/ELAVL3. **d** QPCR showing relative mRNA expression of indicated genes in NCI-H660/shCtrl and NCI-H660/shELAVL3. **e** QPCR showing relative mRNA expression of indicated genes (left) and Western blot showing indicated protein expression (right) in LASCPC-01/shCtrl and LASCPC-01/shELAVL3. **f** QPCR showing relative mRNA expression of indicated genes in shELAVL3 versus shCtrl of LNCaP/AR/shTP53/shRB1 (LAPR). **g** KEGG pathway enrichment analysis of differentially expressed genes from shELAVL3 versus shCtrl in LAPR. **h** Photograph showing LASCPC-01 xenografts in nude mice with shCtrl and two independent shRNAs targeting *ELAVL3*

(left). Tumor volume was measured once a week at indicated time points (middle, *n* = 6 per group). Tumor weight was measured when sacrificed (right, *n* = 6 per group). **i** Representative H&E and immunohistochemistry staining (upper; scale bar, 50 µm) and quantification (bottom; *n* = 9) of indicated staining in LASCPC-01 xenografts. **j** Cell viability of LAPR/shCtrl and LAPR/shELAVL3 treated with 10 µM Enzalutamide at indicated time points (*n* = 3 biologically independent experiments). **k** Cell viability of LNCaP/Ctrl and LNCaP/ELAVL3 treated with 10 µM Enzalutamide at indicated time points (*n* = 3 biologically independent experiments). **l** Cell viability of LNCaP/Ctrl and LNCaP/ELAVL3 treated with increasing concentration of Enzalutamide for 48 h (*n* = 3 biologically independent experiments). Data presented as mean ± s.e.m. (**c–f, h–l**). Statistical significance was determined by one-way ANOVA with Dunnett's multiple comparisons (**e**), or two-tailed unpaired Student's *t*-test (**c, d, f, h–l**). Western blot experiments were repeated three times independently, with similar results (**c, e**). QPCR experiments were conducted *n* = 3 biologically independent experiments. Source data are provided as a Source Data file.

downstream mRNA and that the 3′-UTR of *MYCN* mRNA contains a consensus AU-rich sequence, we reasoned that ELAVL3 may directly bind and regulate *MYCN* mRNA. By analyzing the catRAPID algorithm (http://service.tartaglialab.com/page/catrapid_group), we showed that ELAVL3, rather than other ELAV members, emerged as one of the most detected RNA-binding proteins within the *MYCN* mRNA 3′-UTR, of which the sequences between 1 and 102 being the most possible ELAVL3 binding domain (Supplementary Fig. S4a, Supplementary Table 3).

ELAVL3 is composed of three RNA recognition motifs (RRM1, RRM2, and RRM3), along with a flexible hinge region situated between RRM2 and RRM3, which are highly conserved across different species (Supplementary Table 4). To determine the functional significance of these motifs, we generated a set of ELAVL3 constructs, including a full-length version and deletions that targeted either RRM1, RRM2, RRM3, or the hinge region (Supplementary Fig. S4b). Our results demonstrated that the deletion of neither RRM1 nor RRM2 leads to an increase in *MYCN* mRNA or the development of a neuroendocrine phenotype, suggesting that both RRM1 and RRM2 are crucial for controlling *MYCN* and mediating neuroendocrine differentiation (Fig. 4c). These results were confirmed by the luciferase reporter assay, in which deletion of RRM1 or RRM2 failed to stimulate signals driven by the *MYCN* mRNA 3′-UTR, in contrast to the ELAVL3 wildtype (Fig. 4d). To verify the RNA-binding function of ELAVL3, we performed RNA immunoprecipitation followed by high-throughput sequencing (RIP-seq) in PC3 cells overexpressing ELAVL3-wildtype or ELAVL3-RRM1-deletion. As shown in Fig. 4e, ELAVL3-wildtype, but not ELAVL3-RRM1-deletion, significantly enriched *MYCN* transcripts in RIP-seq peaks. These results by RNA immunoprecipitation-qPCR (RIP-qPCR) showed that *MYCN* mRNA levels were significantly enriched in cells expressing ELAVL3-wildtype, but not in cells expressing ELAVL3-RRM1-deletion (Fig. 4f and Supplementary Fig. S4c). To inspect the specific binding sites in the 3′-UTR of the *MYCN* mRNA, we created luciferase reporter constructs containing full-length *MYCN* mRNA (907 base pairs) and three segments of equal length, each approximately 300 base pairs in length (Fig. 4g). ELAVL3, instead of other members of the ELAV family, markedly stimulated the luciferase signal driven by full-length, P1, and P3, but not so much in P2 (Fig. 4h and Supplementary Fig. S4d). Therefore, we performed an RNA pull-down assay using biotin-labeled RNA probes (F1−6) targeting P1 and P3. As shown in Fig. 4i, ELAVL3 was physically bound to P1 and P3 of the *MYCN* mRNA 3′-UTR in diverse cell lysates. To confirm the direct interaction between ELAVL3 and *MYCN* mRNA, we performed an RNA electrophoretic mobility shift assay (RNA-EMSA) with purified ELAVL3 protein and synthetic RNA probes. Our findings validated the formation of an RNA-protein complex band resulting from the interaction between GST-ELAVL3 and wild-type *MYCN* 3′-UTR probes, but not with non-labeled or mutant probes

(Fig. 4j). These results firmly support the direct binding of ELAVL3 to *MYCN* 3′-UTR mRNA.

Given that ELAVL3 upregulated both the RNA and protein levels of MYCN, we speculated that ELAVL3 might stabilize the mRNA of *MYCN*, a function attributed to the ELAV family of RNA-binding proteins, as reported in a previous study[31]. Therefore, we performed an RNA stability assay in PC3 cells, which stably expressed *MYCN* with or without the 3′-UTR, and added Actinomycin D to block transcription. As shown in Fig. 4k, the presence of ELAVL3 effectively prolonged the half-life of *MYCN* mRNA, indicating that ELAVL3 could protect *MYCN* mRNA from decay. Notably, ELAVL3 moderately prolonged the half-life of *MYCN* mRNA even without the 3′-UTR (Supplementary Fig. S4e). Thus, we cannot exclude the possibility that ELAVL3 also binds to the coding region of *MYCN* mRNA. To further eliminate the influence of pre-existing RNA, we employed a pulse-labeling technique using 4sU to tag mRNA in PC3/*MYCN*−3′-UTR cells and chased for 8 h to detect the degradation rate of mRNA. As shown in Fig. 4l, m, the presence of ELAVL3 extended the half-life of 4sU-labeled *MYCN* mRNA in cells expressing ELAVL3 compared with the vector control. Altogether these data demonstrate that ELAVL3 directly binds to and stabilizes *MYCN* mRNA.

## ELAVL3 binds and stabilizes the mRNA of *RICTOR*

The above KEGG enrichment analysis revealed an upregulation of the PI3K/AKT/mTOR signaling in patient samples with high *ELAVL3* expression (Supplementary Fig. S1d, e). We also observed that ELAVL3 remarkably increased AKT phosphorylation and activated the AKT pathway (Fig. 2c and Supplementary Fig. S2b). These observations prompted us to question whether ELAVL3 regulates AKT/mTOR components. To explore this possibility, we evaluated the mRNA levels of several mTOR components in both LNCaP and PC3 cells and found that *RICTOR* distinguished itself from all other components by showing a significant increase in the presence of ELAVL3 in both cell lines (Fig. 5a and Supplementary Fig. S5a). In contrast, knockdown of ELAVL3 in LASCPC-01 cells significantly reduced both the mRNA and protein levels of RICTOR (Fig. 5b). Indeed, a previous study reported that *RICTOR*, a subunit of mTORC2, harbors conserved motifs in its 3′-UTR that are recognized by RNA-binding proteins and can undergo post-transcriptional regulation[32]. Our survey of RNA-binding protein motifs from catRAPID identified ELAVL3 as one of the most detected RNA-binding proteins in *RICTOR* mRNA 3′-UTR, but not in *MTOR* or *MLST8* (Supplementary Fig. S5b–d). The region between 1504 and 1889 of the *RICTOR* mRNA 3′-UTR was suggested to be the most likely binding domain for ELAVL3 (Supplementary Table 5). We performed RIP-seq analysis and found that *RICTOR* transcripts were enriched in the peaks from cells expressing ELAVL3-wildtype, but not from ELAVL3-RRM1-deletion (Fig. 5c). These results were further

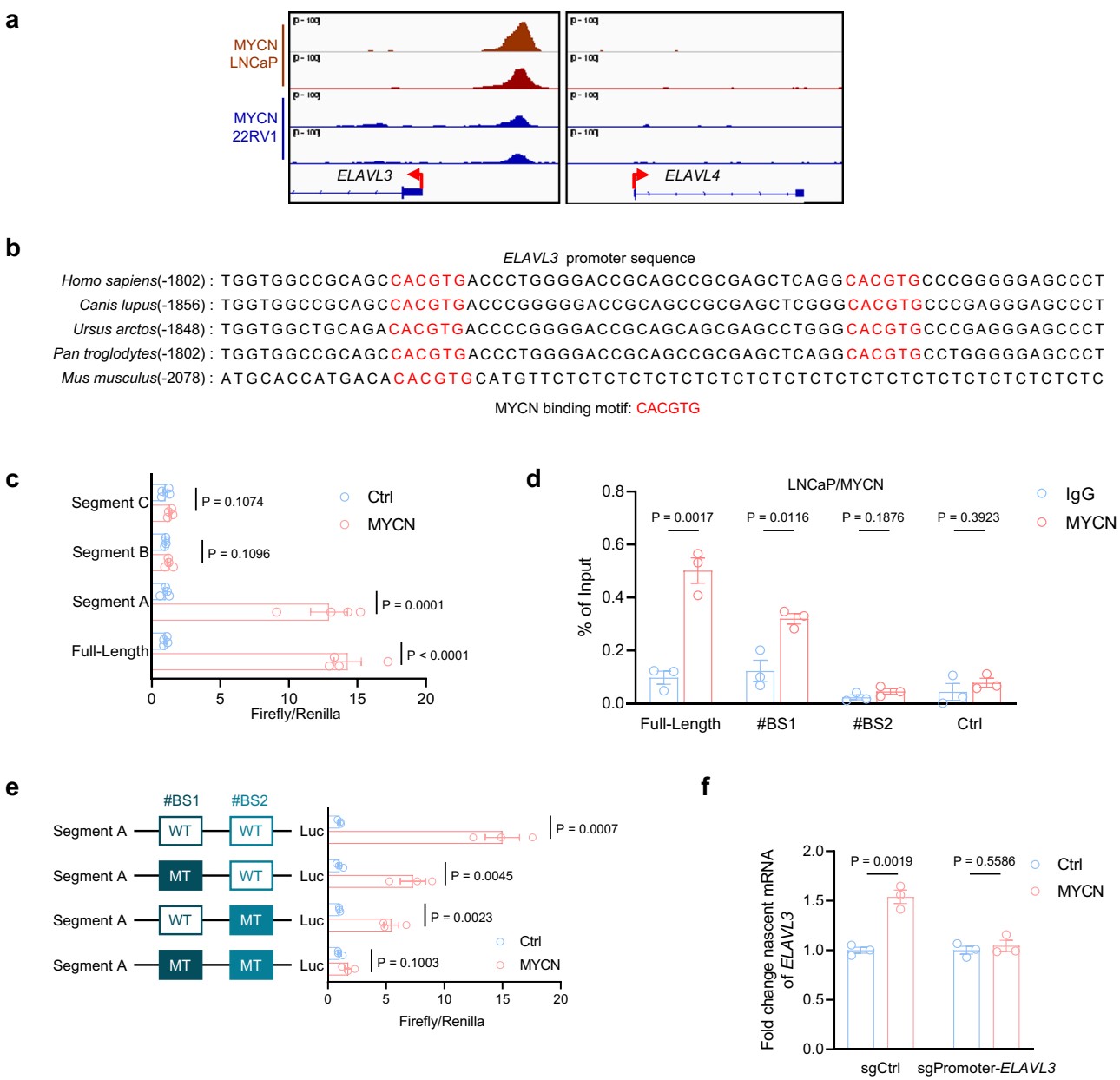

**Fig. 3 | MYCN transcriptionally upregulates ELAVL3 expression. a** ChIP-seq enrichment of MYCN at *ELAVL3* and *ELAVL4* promoter in prostate cancer cell lines (three independent samples; GSE117306[28]). **b** MYCN binding motifs exist on *ELAVL3* promoter across species. **c** Relative luciferase activities driven by full-length or indicated segments of *ELAVL3* promoter in HEK293T (*n* = 3 independently transfected replicates per condition). **d** ChIP-qPCR of different locations on *ELAVL3* promoter from LNCaP/MYCN with either MYCN antibody or IgG (three independent samples). **e** Relative luciferase activities driven by wildtype or indicated

mutant Segment A of *ELAVL3* promoter reporters in HEK293T (*n* = 3 independently transfected replicates per condition). **f** QPCR showing changes of nascent *ELAVL3* mRNA in HEK293T with/without MYCN induction (*n* = 3 biologically independent experiments). The sgRNAs were designed to target the MYCN binding sites in the ELAVL3 promoter region and the non-targeting sgRNA was used as control. Data presented as mean ± s.e.m. (**c**, **d**, **e**, **f**). Statistical significance was determined by two-tailed unpaired Student's *t*-test (**c**, **d**, **e**, **f**). BS, binding site. Source data are provided as a Source Data file.

confirmed by RIP-qPCR (Fig. 5d and Supplementary Fig. S5e), indicating direct binding of ELAVL3 to *RICTOR* mRNA. In addition, the luciferase reporter assay demonstrated that both ELAVL3 wildtype and deletion mutants of RRM3 or hinge could enhance the luciferase signal driven by the full-length *RICTOR* mRNA 3'-UTR. In contrast, the deletion of either RRM1 or RRM2 failed to produce the same effect, indicating the essential role of functional RRM1 and RRM2 domains in RNA-binding activity, which is required for the activation of the luciferase assay (Fig. 5e). To inspect the specific binding motifs in the 3'-UTR mRNA of *RICTOR*, we performed an RNA pull-down assay using biotin-labeled RNA probes (F1–6). These probes were designed to target the region between 1504 and 1889 in the 3'-UTR, which were

predicted to be the most likely binding sites for ELAVL3 (Supplementary Fig. S5f). As shown in Fig. 5f, ELAVL3 was physically bound to several sites in the 3'-UTR of *RICTOR* mRNA in various cell lysates. To conduct a more detailed examination of the interaction between ELAVL3 and *RICTOR* mRNA, we performed RNA-EMSA with purified ELAVL3 protein and synthetic RNA probes designed to target the 3'-UTR of *RICTOR*. Our findings demonstrated the formation of an RNA-protein complex band arising from the interaction of GST-ELAVL3 and the wild-type *RICTOR* 3'-UTR probes. This complex formation was not observed with non-labeled or mutant probes (Fig. 5g). Additionally, we generated luciferase reporter constructs containing full-length (4384 bp) and four distinct segments of the *RICTOR* mRNA

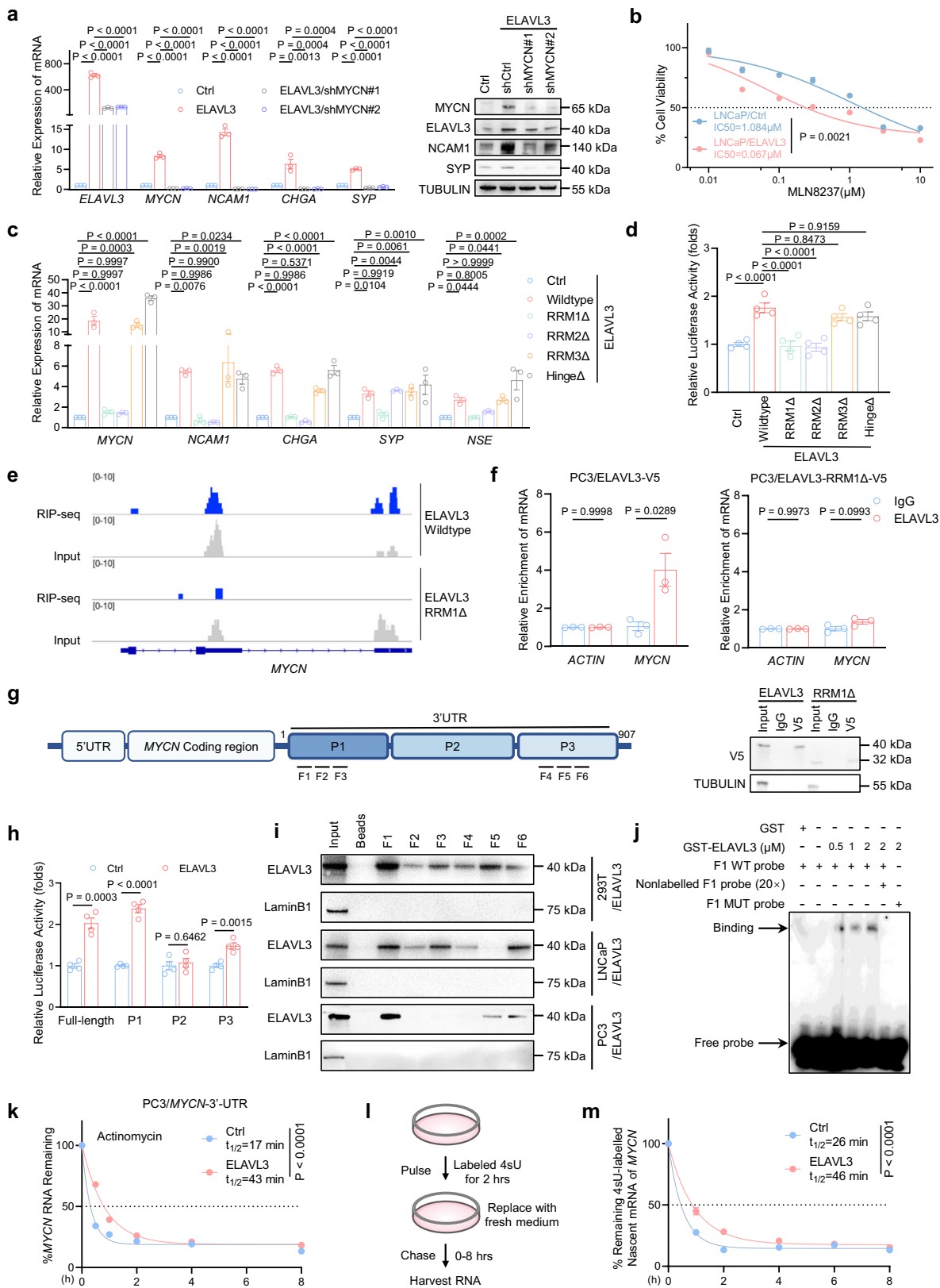

3′-UTR (Segment S1, S2, S3, and S4). As shown in Fig. 5h, wild-type ELAVL3 effectively activated the luciferase signal driven by all four separated segments of *RICTOR* mRNA 3′-UTR, whereas the RRM1 deletion construct did not show similar activation. These results provide robust evidence supporting the direct binding of ELAVL3 to the *RICTOR* 3′-UTR mRNA.

Next, we performed an RNA stability assay in LASCPC-01 cells using Actinomycin D to block transcription. As shown in Fig. 5i, the overexpression of ELAVL3 prolonged the half-life of *RICTOR* mRNA, whereas the knockdown of ELAVL3 resulted in a shorter half-life. In contrast, ELAVL3 did not significantly alter the stability of *MTOR* mRNA (Supplementary Fig. S5g). Moreover, pulse-chasing of 4sU-labeled

**Fig. 4 | ELAVL3 binds and stabilizes the mRNA of *MYCN*. a** QPCR showing relative mRNA expression of indicated genes (left) and Western blot showing indicated protein expression (right) in indicated LNCaP cell lines. **b** Cell viability of LNCaP/Ctrl and LNCaP/ELAVL3 treated with increasing concentration of MLN8237 for 48 h (*n* = 3 biologically independent experiments). **c** QPCR showing relative mRNA expression of indicated genes in PC3 expressing indicated constructs of ELAVL3. **d** Relative luciferase activities driven by full-length of *MYCN* mRNA 3′-UTR reporters in HEK293T transfected with indicated constructs of ELAVL3 (*n* = 4 independently transfected replicates). **e** Genome browser tracks showing representative ELAVL3-wildtype and ELAVL3-RRM1-deletion RIP-seq peaks at the *MYCN* gene.
**f** Ribonucleoprotein immunoprecipitation (RIP) assay using either V5 antibody or IgG showing the relative enrichment of *MYCN* mRNA in PC3/ELAVL3-wildtype and PC3/ELAVL3-RRM1-deletion (upper, *n* = 3 biologically independent experiments). Western blot showing ELAVL3 was pulled down by V5 antibody (bottom).
**g** Schematic diagram showing putative ELAVL3 binding sites on *MYCN* mRNA 3′-UTR, segments (P1-P3) designed for luciferase reporter assay, and fragments (F1-F6) designed for RNA pull-down assays. **h** Relative luciferase activities driven by full-length or indicated segments of *MYCN* mRNA 3′-UTR reporters in HEK293T transfected with ELAVL3 or empty vector (*n* = 4 independently transfected replicates). **i** Western blot showing ELAVL3 expression in pull-downed lysates by indicated biotinylated RNA probes (F1-F6) in indicated cells. **j** RNA-EMSA showing the RNA-protein complex bands of purified GST-ELAVL3 protein and indicated RNA probes (*n* = 3 biologically independent experiments). **k** *MYCN* mRNA decay in PC3/MYCN-3′-UTR with/without ELAVL3 expression at indicated time points following 5 μg/mL Actinomycin D treatment (*n* = 3 biologically independent experiments). **l** Schematic diagram showing pulse-labeling of nascent RNA procedure. **m** RNA decay of 4sU-labeled nascent *MYCN* mRNA in PC3/MYCN-3′-UTR with/without ELAVL3 expression (*n* = 3 biologically independent experiments). Data presented as mean ± s.e.m. (**a**–**d**, **f**, **h**, **k**, **m**). Statistical significance was determined by one-way ANOVA with Dunnett's multiple comparisons (**a**, **c**), or one-way ANOVA with Sidak's multiple comparisons (**d**), or two-tailed unpaired Student's *t*-test (**b**, **f**, **h**, **k**, **m**). Western blot experiments were repeated three times independently, with similar results (**a**, **f**, **i**, **j**). QPCR experiments were conducted *n* = 3 biologically independent experiments. Δ, deletion. Source data are provided as a Source Data file.

nascent RNA indicated an increase in *RICTOR* mRNA stability in LASCPC-01 cells with ELAVL3 overexpression and a reduction in knockdown cells (Supplementary Fig. S5h). These results suggested that ELAVL3 directly binds to and stabilizes *RICTOR* mRNA.

Recent research has demonstrated an association between copy number alterations and activation of members of the AKT signaling pathway, including *PTEN*, *PIK3CA*, *RICTOR*, and *MTOR*, with neuroendocrine differentiation in prostate cancer[33,34]. Our previous results suggested that depletion of ELAVL3 led to a striking reduction in *RICTOR* (Fig. 5b). Consistently, we found that both the knockdown of *RICTOR* and pharmacological inhibition of the AKT pathway with MK-2206 effectively suppressed neuroendocrine differentiation in ELAVL3 overexpression cells (Fig. 5j, k and Supplementary Fig. S5i, j). Collectively, our results suggest that ELAVL3 may be responsible for promoting neuroendocrine differentiation of prostate cancer cells by stabilizing *RICTOR* mRNA.

## Pharmacological inhibition of ELAVL3 blocks neuroendocrine differentiation of prostate cancer cell

These above findings support the pivotal role of the interplay between ELAVL3 and MYCN in driving neuroendocrine differentiation in prostate cancer. This prompted us to explore potential strategies for disrupting this interaction. Previous studies have reported various compounds with the capacity to modulate the functions of ELAV family proteins. To identify potential inhibitors targeting the ELAVL3-MYCN interaction, we subjected a panel of compounds to rigorous testing for possible inhibitory effects on the expression of *ELAVL3*, *MYCN*, and neuroendocrine markers in LASCPC-01 cells. As shown in Fig. 6a, pyrvinium pamoate (PP) stood out by demonstrating the most robust inhibitory effect among all candidates at the same concentration and duration of treatment (100 nM, 24 and 48 h). Cell viability assay showed that 100 nM PP decreased the viability of LASCPC-01 cells by 80% after 72 h, further confirming its potency as an inhibitor (Fig. 6b). Notably, treatment with 100 nM PP resulted in reduced levels of MYCN, ELAVL3, RICTOR, and neuroendocrine markers while simultaneously inducing cell apoptosis, as measured by cleaved-PARP, across multiple time points (Fig. 6c).

Pyrvinium pamoate (PP) is an FDA-approved anthelmintic that inhibits the formation of the ELAVL1-RNA complex by disrupting its cytoplasmic translocation[35]. Owing to the high protein sequence homology between ELAVL1 and ELAVL3, we reasonably expected that PP may inhibit the function of ELAVL3, thereby impeding the neuroendocrine differentiation of prostate cancer cells. As shown in Fig. 6d, the knockdown of ELAVL3 reduced the sensitivity of LASCPC-01 cells to PP treatment, indicating that ELAVL3 may be the target of PP-mediated cell death. To test whether PP alters the subcellular localization of ELAVL3, we prepared cytoplasmic and nuclear extracts

from LASCPC-01 cells and analyzed the endogenous level of ELAVL3. As shown in Supplementary Fig. S6a, PP treatment did not affect the cytoplasmic to nuclear expression ratio of ELAVL3. Moreover, we overexpressed GFP-fused ELAVL3 in PC3 cells and performed immunofluorescence staining to examine the localization of ELAVL3 with or without PP treatment. Consistently, PP failed to change the nucleo-cytoplasmic shuttling on ELAVL3 (Supplementary Fig. S6b). Since the expression of *ELAVL3*, *MYCN*, and *RICTOR* was reduced by PP treatment, we investigated whether PP could disrupt the interaction between ELAVL3 and its target RNAs. To test this hypothesis, we first performed a luciferase reporter assay driven by the full-length *MYCN* mRNA 3′-UTR. In HEK293T cells, 200 nM PP was sufficient to reverse the ELAVL3-mediated elevation of the luciferase signal (Fig. 6e). RIP-qPCR also demonstrated that PP effectively reduced the enrichment of *MYCN* mRNA 3′-UTR in the ELAVL3-immunoprecipitated complex in both LNCaP and PC3 cells (Fig. 6f and Supplementary Fig. S6c). To test the direct disruption of ELAVL3 function by PP, we performed an RNA-EMSA assay and showed that PP treatment effectively reduced the formation of RNA-protein complex bands in vitro (Fig. 6g). These results suggest that PP inhibits ELAVL3 function by interrupting its binding to the mRNA.

Moreover, we performed RNA-seq on NCI-H660 cells treated with PBS or PP. Gene Set Enrichment Analysis suggested that PP treatment significantly suppressed the oncogenic traits, including the mTORC1, Hedgehog, P53, TNFα/NFκB, and MYC pathways, consistent with our previous RNA-sequencing and RIP-seq analyses on different cell lines (Fig. 6h and Supplementary Fig. S6d). To test the efficacy of PP in vivo, DU145 cells stably expressing ELAVL3 were injected subcutaneously into nude mice. Once the tumor reached approximately 50 mm³, the mice were randomly assigned to either the PP treatment or the PBS control group (*n* = 5 per group). We showed that PP treatment dramatically suppressed tumor growth by measuring the tumor volume and weight (Fig. 6i). Immunohistochemical staining revealed that PP treatment reduced the expression of ELAVL3 and Ki67, when compared with that in the PBS group (Fig. 6j, k). These findings demonstrated that PP effectively inhibited ELAVL3 by disrupting mRNA binding and that PP could reduce tumor growth in NEPC.

## Extracellular vesicles-mediated transfer of ELAVL3 induces neuroendocrine differentiation in recipient cancer cells

Recent studies have reported that NEPC cells can induce neuroendocrine differentiation of adenocarcinoma cells in an extracellular vesicle-mediated manner[36–38]. A similar phenotype has been observed in ELAVL3-expressing prostate cancer cells. In our experiments, we cultured prostate cancer cells (PC3, DU145, and LNCaP) that either expressed ELAVL3-wildtype or vector control for 48 h, and collected the supernatants as the conditioned medium. We then treated the

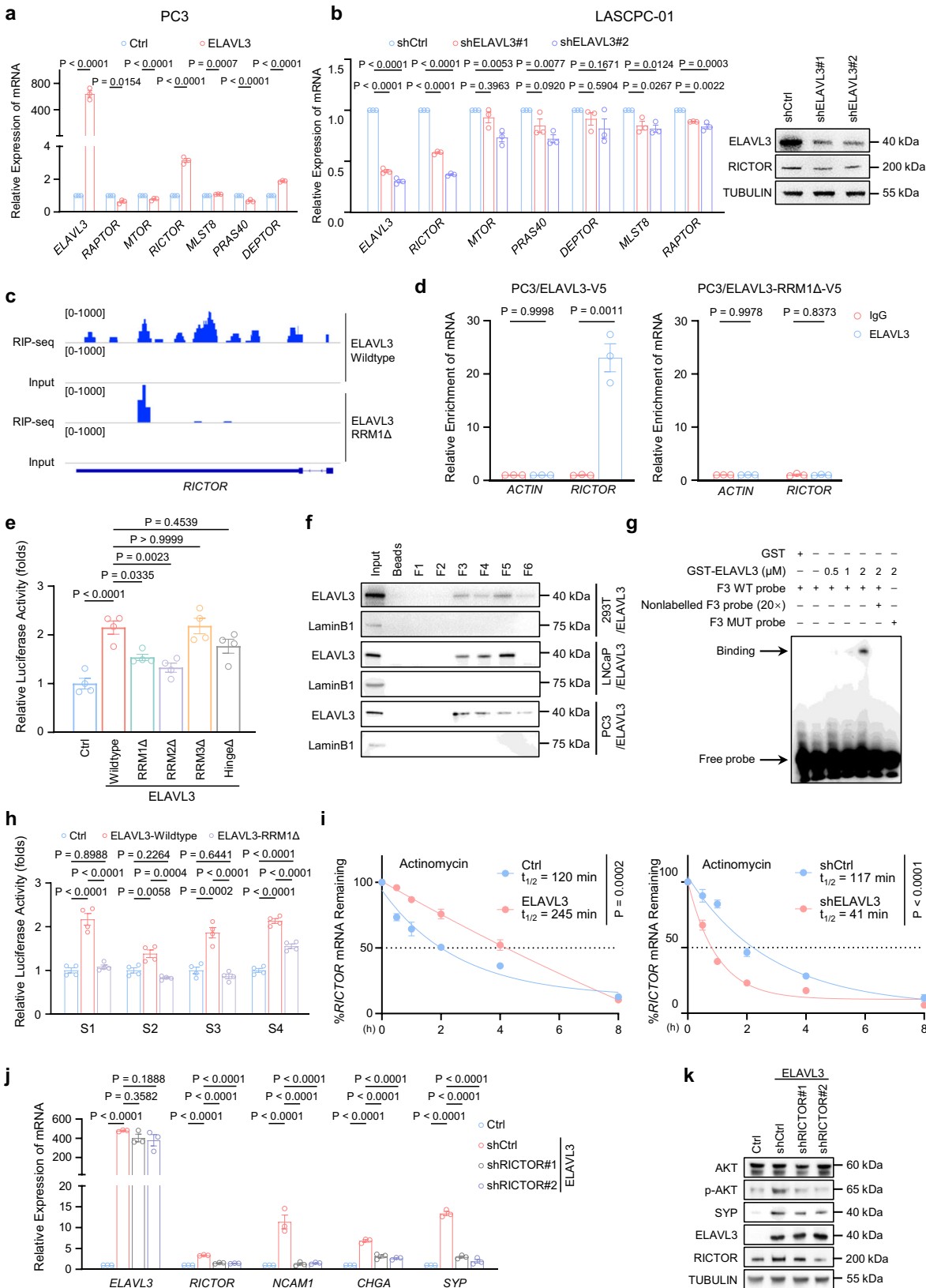

recipient cells (PC3, DU145, and LNCaP) with the above conditioned medium and found that recipient cells treated with the medium derived from cells expressing ELAVL3, but not from vector control, showed upregulation of neuroendocrine markers, along with an increase in *ELAVL3* and *MYCN* (Fig. 7a and Supplementary Fig. S7a). We isolated and characterized EVs from the conditioned medium using transmission electron microscopy, Nanoparticle Flow Cytometry, and iodixanol/Optiprep density gradient (Supplementary Fig. S7b–d). Western blot analysis confirmed the presence of the EVs markers ALIX and TSG101, whereas the endoplasmic reticulum marker CALNEXIN was absent (Fig. 7b). Notably, both the RNA and protein of ELAVL3 were detected in the EVs derived from PC3 cells expressing ELAVL3,

**Fig. 5 | ELAVL3 binds and stabilizes the mRNA of *RICTOR*. a** QPCR showing relative mRNA expression of *ELAVL3* and *mTORC2*-related genes in indicated PC3 cells. **b** QPCR (left) and Western blot (right) showing indicated genes in LASCPC-01 with control shRNA and two independent shRNA targeting ELAVL3. **c** Genome browser tracks showing representative ELAVL3-wildtype and ELAVL3-RRM1-deletion RIP-seq peaks at the *RICTOR* gene. **d** RIP assay using either ELAVL3 antibody or IgG showing the relative enrichment of *RICTOR* mRNA in PC3/ELAVL3-wildtype and PC3/ELAVL3-RRM1-deletion (n = 3 biologically independent experiments). **e** Relative luciferase activities driven by full-length of *RICTOR* mRNA 3′-UTR reporters in HEK293T transfected with indicated ELAVL3 or control (n = 4 independently transfected replicates). **f** Western blot showing ELAVL3 expression in pull-downed lysates by indicated biotinylated RNA probes (F1-F6) in indicated cells. **g** RNA-EMSA showing the RNA-protein complex bands of purified GST-ELAVL3 protein and indicated RNA probes (n = 3 biologically independent experiments). **h** Relative luciferase activities driven by segments S1, S2, S3, and S4 of *RICTOR*

mRNA 3′-UTR reporters in HEK293T transfected with indicated ELAVL3 or control (n = 4 independently transfected replicates). **i** *RICTOR* mRNA decay in LASCPC-01/Ctrl and LASCPC-01/ELAVL3 at indicated time points post-treatment with 5 μg/mL Actinomycin D (left). *RICTOR* mRNA decay in LASCPC-01/shCtrl and LASCPC-01/shELAVL3 at indicated time points post-treatment with 5 μg/mL Actinomycin D (right) (n = 3 biologically independent experiments). **j** QPCR showing the level of *ELAVL3*, *RICTOR*, and NE-related genes (*NCAM1*, *CHGA*, and *SYP*) in indicated PC3 cells. **k** Western blot showing the level of ELAVL3, RICTOR, SYP, p-AKT, and AKT in indicated PC3 cells. Data presented as mean ± s.e.m. (**a, b, d, e, h, i, j**). Statistical significance was determined by one-way ANOVA with Dunnett's multiple comparisons (**b, h, j**), or one-way ANOVA with Sidak's multiple comparisons (**e**), or two-tailed unpaired Student's *t*-test (**a, d, i**). Western blot experiments were repeated three times independently, with similar results (**b, c, f**). QPCR experiments were conducted n = 3 biologically independent experiments. Δ, deletion. Source data are provided as a Source Data file.

but not in those derived from the vector control, indicating the release of ELAVL3 into these EVs (Fig. 7b, c). We further confirmed the presence of ELAVL3 in EVs by using PC3 cells expressing GFP-fused ELAVL3. As shown in Fig. 7d, nanoflow cytometry analysis captured the GFP fluorescence in the EVs derived from cells expressing ELAVL3-GFP, but not from the non-GFP control, suggesting that ELAVL3 protein was secreted into EVs. Next, we labeled the isolated EVs with the liquid fluorescent dye PKH67 and incubated them with PC3 cells for 6 h. Fluorescence microscopy detected a punctate signal of PKH67 inside the cytoplasm of recipient PC3 cells, indicating the internationalization of these EVs (Fig. 7e).

Next, we isolated EVs from the conditioned medium of PC3 cells expressing wild-type ELAVL3, RRM1-deletion ELAVL3, or vector control. After treatment with proteinase K, EVs were co-cultured with recipient PC3 cells for 48 h. As shown in Supplementary Fig. S7e, only EVs from wild-type ELAVL3 cells induced neuroendocrine markers (*CHGA* and *SYP*) in recipient cells, concomitant with an increase in *ELAVL3* and *MYCN*. Notably, PP treatment effectively eliminated the EVs-mediated effects in recipient cells (Fig. 7f). Finally, we isolated serum EVs from patients with hormone-sensitive prostate cancer or neuroendocrine prostate cancer and found a significant increase in *ELAVL3* and *MYCN* mRNA levels in the NEPC samples (Fig. 7g). However, we did not observe this increase in ELAVL3 protein levels, which may be due to trace amounts in the patient sera (Supplementary Fig. S7f). Taken together, these data suggest that, in addition to the intracellular manner, ELAVL3 can be released in EVs and induce neuroendocrine differentiation of recipient adenocarcinoma cells, thereby propagating the phenotype via an intercellular mechanism.

### Pharmacological inhibition of ELAVL3 suppresses tumor growth and promotes survival in NEPC mouse model

Given the encouraging results of PP's ability to suppress tumor growth and prevent the propagation of neuroendocrine phenotype, we investigated its potential effectiveness as a treatment in a genetically engineered mouse model of neuroendocrine prostate cancer. We examined the efficacy of PP treatment on organoids generated from two genetically engineered mouse models of prostate cancer, namely Pb-Cre4: *Pten*^f/f; *Trp53*^f/f (DKO) and Pb-Cre4: *Pten*^f/f; *Trp53*^f/f; *Rb1*^f/f (TKO) (Supplementary Fig. S8a). As shown in Fig. 8a, b, the inhibitory effect of PP on organoids from TKO mice was stronger than that on organoids from DKO mice, as evidenced by measurements of viability and colony formation. Immunofluorescence analysis revealed that PP treatment significantly reduced the expression of Elavl3 and Syp in the TKO organoids (Supplementary Fig. S8b). To determine the anti-tumor effect of PP in vivo, TKO mice were surgically castrated at 9–10 weeks of age and randomly assigned to receive intraperitoneal injection of PP 0.2 mg/kg or PBS every other day for four weeks. Mice underwent [^18F]-FDG PET/CT before the first treatment and again after four weeks of PP therapy (Fig. 8c). As shown in Fig. 8d, e, PP treatment effectively

reduced the [^18F]-FDG PET/CT signal, quantified by the maximum standardized uptake value, when compared with the PBS group, despite both groups exhibiting comparable baseline [^18F]-FDG PET/CT signals. Importantly, the median survival for mice treated with PP was 122 days, compared with 107 days for those receiving PBS (P = 0.0005, Fig. 8f). Furthermore, we assessed the effect of PP on tumor proliferation using EdU immunofluorescence staining and found that PP effectively reduced the EdU-positive ratio and Elavl3 expression (Fig. 8g, h). Notably, such EdU-positive cells overlapped with residual Elavl3 expression, again indicating that the residual Elavl3 was required for cell proliferation. The subsequent histopathological analysis revealed that PP treatment significantly reduced the proportion of Ki67-positive cells and induced cleaved caspase-3 positive cells, indicating that PP treatment caused a reduction in cell proliferation and induction in apoptosis (Fig. 8i, j). Meanwhile, we observed a significant decrease in tumor metastatic burden in the PP treatment group when compared with the PBS control by quantifying the number of metastases in both the liver and lung (Fig. 8k, l). We collected tumor tissues from the mice and measured the RNA and protein levels of Elavl3. As shown in Fig. 8m, n, PP treatment effectively reduced Elavl3 expression in tumor samples.

Given the controversial use of [^18F]-FDG PET/CT and the potential interference from bladder tracer accumulation, we developed another NEPC mouse model by surgically injecting a lentiviral Luc.Cre reporter into the prostate of *Pten*^f/f; *Trp53*^f/f; *Rb1*^f/f mice. This allowed us to track tumor burden through bioluminescence imaging[39]. After measuring the baseline luciferase signal 9–10 weeks after Luc.Cre injection, we randomly assigned the mice into two groups and administered either PP or PBS every other day for 8 weeks. Luciferase signals were measured again after 8 weeks of PP therapy (Supplementary Fig. S8c). Our analysis revealed that the PP treatment caused a considerable reduction in the luciferase signal compared with the PBS group, although there was no distinction in the base level of the signal between the two groups (Supplementary Fig. S8d, e). No significant weight loss was observed in any of these mouse models, indicating that long-term PP treatment does not cause severe toxicity (Supplementary Fig. S8f). Collectively, these data strongly suggest that PP can inhibit tumor growth and prolong the survival of mice with NEPC.

In summary, our study established a critical role of ELAVL3 in maintaining the neuroendocrine phenotype of prostate cancer. We found that *ELAVL3* is transcriptionally activated by MYCN, and subsequently binds to and stabilizes the mRNA of both *MYCN* and *RICTOR*. ELAVL3 can be released into EVs and induce neuroendocrine differentiation in recipient cells. Additionally, our investigations demonstrated that PP disrupts the interaction between ELAVL3 and downstream mRNA, resulting in the reduction of ELAVL3-induced neuroendocrine differentiation and suppression of tumor growth. This discovery provides a promising strategy for drug repurposing of the treatment of NEPC (Fig. 8o).

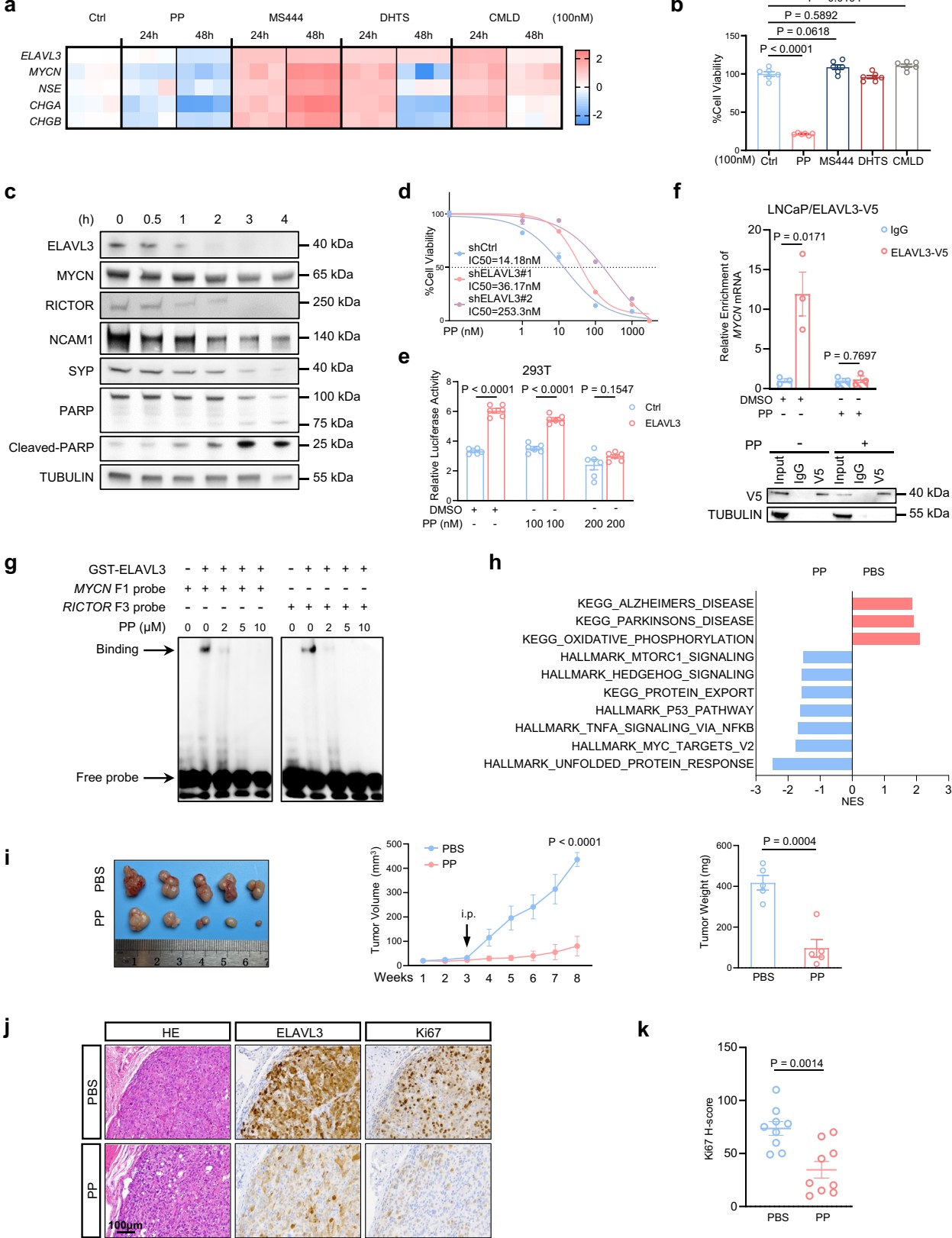

## Discussion

Effective treatment of NEPC patients remains a clinical challenge, emphasizing the need to identify targetable oncogenic drivers for developing potential therapeutic strategies. In this study, we identified ELAVL3 as a critical regulator of neuroendocrine differentiation and provided mechanistic insights into the mutual regulation of ELAVL3 and MYCN in the pathogenesis of NEPC. We discovered that *ELAVL3* is transcriptionally activated by MYCN, and subsequently binds to and stabilizes the mRNA of both *MYCN* and *RICTOR*. This interaction gives rise to a positive feedback loop that promotes neuroendocrine differentiation and enhances tumor growth. The ELAV family of RNA-binding proteins has gained significant attention due to their potential

**Fig. 6 | Pharmacological inhibition of ELAVL3 blocks neuroendocrine differentiation of prostate cancer cell. a** Heatmap showing the relative mRNA expression of *ELAVL3*, *MYCN*, and NE-related genes in LASCPC-01 with indicated treatments. Data are normalized to 0 h. ($n = 3$ biologically independent experiments). **b** Cell viability of LASCPC-01 with indicated treatments ($n = 6$ biologically independent experiments). **c** Western blot showing the expression of ELAVL3, MYCN, RICTOR, NE-related genes, and apoptosis markers (PARP and cleaved-PARP) in LASCPC-01 treated with 100 nM PP at indicated time points. **d** Cell viability of LASCPC-01/shCtrl and LASCPC-01/shELAVL3 treated with increasing concentration of PP ($n = 3$ biologically independent experiments). **e** Relative luciferase activities driven by *MYCN* mRNA 3'-UTR reporters in HEK293T treated with PP or DMSO ($n = 6$ independently transfected replicates). **f** RIP assay using either ELAVL3 antibody or IgG showing the relative enrichment of *MYCN* mRNA in LNCaP/ELAVL3 treated with 100 nM PP or DMSO (upper, $n = 3$ biologically independent experiments). Western blot showing ELAVL3 was pulled down by V5 antibody (bottom). **g** RNA-EMSA showing the RNA-protein complex bands of purified GST-ELAVL3 protein and RNA probes with/without PP treatment ($n = 3$ biologically independent experiments). **h** Hallmark gene sets and KEGG pathway enrichment analysis of different expressed genes from NCI-H660 treated with PP versus with PBS. RNA-seq was performed on three independent samples. **i** Photograph showing xenografts of DU145/ELAVL3 from nude mice treated with PP or PBS (left, $n = 5$ per group; scale bar, 1 cm). Tumor volume was measured once a week at indicated time points (middle). Tumor weight was measured when sacrificed (right, $n = 5$ per group). **j** Representative H&E and indicated immunohistochemistry staining in DU145/ELAVL3 xenografts treated with PP or PBS. Scale bar, 100 μm. **k** Quantification of Ki67 staining in DU145/ELAVL3 xenografts treated with PP or PBS ($n = 10$ sections from each mouse used for statistical analysis). Data presented as mean ± s.e.m. (**b, d, e, f, i, k**). Statistical significance was determined by one-way ANOVA with Dunnett's multiple comparisons (**b**), or two-tailed unpaired Student's *t*-test (**e, f, i, k**). Western blot experiments were repeated three times independently, with similar results (**c, f**). PP, pyrvinium pamoate. DHTS, Dihydrotanshinone I. Source data are provided as a Source Data file.

as therapeutic targets for cancer treatment, as well as their ability to regulate a wide range of RNA targets post-transcriptionally[40–42]. Previous studies have reported that ELAVL4 binds to the 3'-UTR of *MYCN* mRNA in neuroblastoma[43–45]. However, our analysis found that ELAVL3 was highly expressed in NEPC through the combination of multiple RNA-seq datasets from prostate cancer patients (Fig. 1a). Furthermore, we have demonstrated that ELAVL3, but not other members of the ELAV family, binds to the *MYCN* 3'-UTR and protects *MYCN* from RNA decay (Fig. 4d, m, and Supplementary Fig. S4d, e). These findings enhance the understanding of the post-transcriptional regulation of *MYCN*.

In addition to the aforementioned intracellular mechanism, we demonstrated that ELAVL3 can be released into EVs, thereby inducing neuroendocrine differentiation in recipient cells. These findings are congruent with previous studies on EV-mediated cell-cell communication between neuroendocrine prostate cancer cells and adenocarcinoma cells[36,38,46]. We also demonstrated that both the mRNA and protein of ELAVL3 were detectable in the EVs derived from ELAVL3-expressed PC3 cells. However, only the mRNA of *ELAVL3* was detectable in EVs derived from NEPC patients. While the Western blot did not reveal detectable levels of ELAVL3 protein, it is possible that ELAVL3 protein was present in trace amounts in the patient sera (Supplementary Fig. S7f). Interestingly, previous studies have reported the presence of ELAVL3 autoantibodies in sera derived from small cell lung cancer patients[47–49]. It is evident from these studies and our own that ELAVL3 holds promise as a potential serum biomarker for the diagnosis of neuroendocrine tumors in the future.

In contrast, the interaction between MYCN and ELAVL3 has provided a potential therapeutic vulnerability amenable to the treatment of NEPC. We evaluated multiple ELAV inhibitors that were previously reported and found that PP was the most effective in hindering ELAVL3 function, compared with the others. Our findings contradict those of previous studies that reported the direct or indirect inhibitory effect of PP on cytoplasmic accumulation of ELAVL1 in bladder cancer cells[35,50]. Instead, we demonstrated that PP treatment did not affect ELAVL3 nuclear-cytoplasmic shuttling, but disrupted the interaction of ELAVL3 and its downstream mRNA, herein *MYCN* and *RICTOR*, subsequently leading to a reduction in ELAVL3-induced neuroendocrine differentiation and concomitant suppression of tumor growth. However, we cannot exclude the possibility of other targets or mechanisms of PP in this process. Indeed, PP showed efficacy in inhibiting NEPC tumor growth not only in the LASCPC-01 xenograft model characterized by robust MYCN expression but also in TKO mice characterized by a mild degree of MYCN expression. Previous studies have reported that PP affects the PI3K/AKT and WNT pathways, both of which play crucial roles in the pathogenesis of neuroendocrine tumors[51,52]. Therefore, it is reasonable to expect that PP may simultaneously affect multiple pathways

during NEPC therapy. Our study expands the scope of the application of PP, which was originally an FDA-approved anthelmintic drug, for the treatment of NEPC. This provides a promising strategy for drug repurposing for further clinical investigation.

Our study has some limitations, one of which pertains to the lack of a patient-derived model. Although we carried out in vitro experiments on human cancer cell lines, we did not evaluate the efficacy of PP on patient-derived organoids or xenografts, which could better reflect the biological and clinical potential of patient tumors. Thus, future research is necessary to strengthen the therapeutic potential of PP in NEPC.

In conclusion, our study identified ELAVL3 as a pivotal oncogenic driver in the initiation and maintenance of neuroendocrine differentiation in prostate cancer. We also elucidated a positive feedback loop involving ELAVL3 and MYCN and highlighted the clinical potential of ELAVL3 in the diagnosis and targeted therapy of patients with NEPC.

## Methods

### Study approval

All the procedures involving mice in this study were approved by Shanghai Jiao Tong University School of Medicine, Renji Hospital Ethics Committee (approval numbers: RA-2020-249). The collection of human samples and research conducted in this study was approved by the Research Ethics Committee of the Renji Hospital, Shanghai Jiao Tong University School of Medicine (approval numbers: KY2022-136A). Clinical samples and information were collected after written informed consent. All animal experiments were performed in compliance with the Guide for the Care and Use of Laboratory Animals (National Academies Press, 2011) and were approved by the Ren Ji Hospital Laboratory Animal Use and Care Committee. This study is compliant with the "Guidance of the Ministry of Science and Technology (MOST) for the Review and Approval of Human Genetic Resources", which requires formal approval for the export of human genetic material or data from China.

### Cell culture

The human prostate cancer cell lines LNCaP (SCSP-5021), 22Rv1 (SCSP-5022), PC3 (SCSP-532), DU145 (SCSP-5024), and VCaP (SCSP-5034)cells were purchased from the Cell Bank, Shanghai Institutes for Biological Sciences, Chinese Academy of Sciences, the LAPC4 cell line was provided by Professor Charles L. Sawyers, cultured in RPMI-1640 medium or DMEM (Gibco) supplemented with 10% fetal bovine serum (Invitrogen), 100 U/mL penicillin, and 100 μg/mL streptomycin (Gibco). LASCPC-01 (CRL-3356) and NCI-H660 (CRL-5813) cells were purchased from ATCC, cultured in HITES medium, supplemented with 5% fetal bovine serum (Invitrogen), 100 U/mL penicillin, and 100 μg/mL streptomycin. All the cells were incubated at 37 °C in a humidified incubator with 5% $CO_2$ and routinely tested for mycoplasma. The

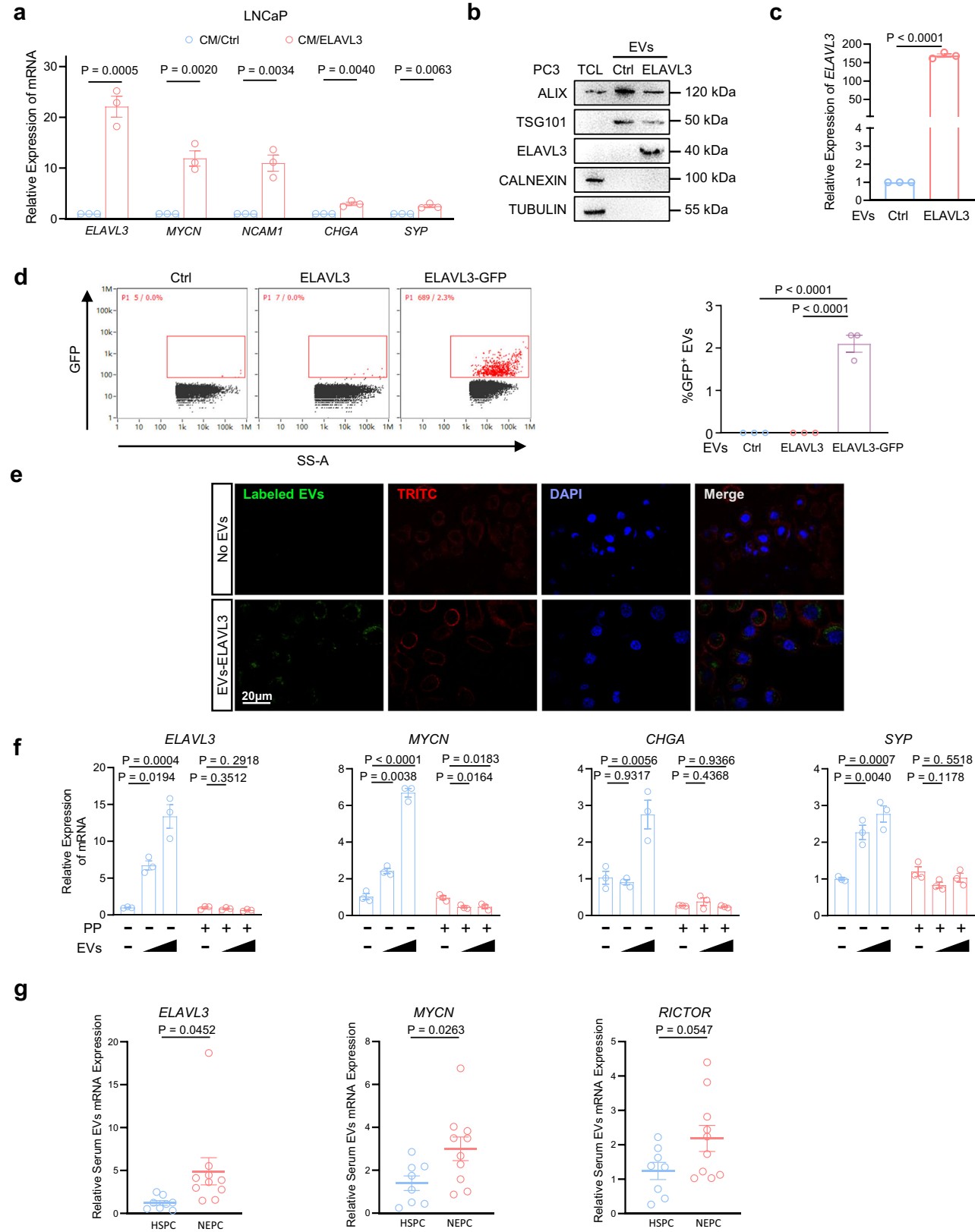

identity of the cell lines was verified through high-resolution small tandem repeats (STR) profiling.

## Western blot

Cells, tissues, and extracellular vesicles were lysed in 1% SDS lysis buffer (P0013G, Beyotime, China), and the protein concentration was determined by BCA Protein Assay (23225, Thermo Fisher Scientific). Equal amounts of protein were separated by 10% SDS-PAGE gel and transferred to 0.45 μm PVDF membranes (Millipore). The membranes were blocked with 5% BSA in TBST and incubated with specific antibodies at 4 °C overnight. Appropriate secondary antibodies were then used, and an ECL detection system (Tanon, China) was used to detect

**Fig. 7 | Extracellular vesicles-mediated transfer of ELAVL3 induces neuroendocrine differentiation in recipient cancer cells. a** QPCR showing relative mRNA expression of NE-related genes (*MYCN*, *NCAM1*, *CHGA*, and *SYP*) in LNCaP treated with indicated conditioned medium (*n* = 3 biologically independent experiments). **b** Western blot showing protein expression of ELAVL3, the EVs markers (ALIX and TSG101), the reticular marker (CALNEXIN), and TUBULIN in total cell lysate (TCL) or in EVs purified from indicated PC3 cells. **c** QPCR showing relative mRNA expression of *ELAVL3* in EVs purified from PC3 expressing ELAVL3 or vector control (*n* = 3 biologically independent experiments). **d** Flow NanoAnalyzer analysis showing the fluorescence signal of EVs purified from indicated PC3 cells (left). Quantification of fluorescent signal of Flow NanoAnalyzer analysis from indicated EVs (right, *n* = 3 biologically independent experiments). **e** Confocal immunofluorescence showing PC3 cells cultured with/without PKH26 labeled EVs purified

from PC3/ELAVL3 (*n* = 3 biologically independent experiments). Scale bar, 20 μm. **f** QPCR showing relative mRNA expression of indicated genes in PC3 cultured with EVs (0, 1 × 10$^{10}$, and 2 × 10$^{10}$) and treated with 100 nM PP or PBS (*n* = 3 biologically independent experiments). **g** QPCR showing relative mRNA expression of *ELAVL3*, *MYCN*, and *RICTOR* in EVs purified from indicated patient groups (*n* = 8 HSPC samples, *n* = 10 NEPC samples). Data presented as mean ± s.e.m. (**a, c, d, f, g**). Statistical significance was determined by two-tailed unpaired Student's *t*-test (**a, c, g**), or one-way ANOVA with Sidak's multiple comparisons (**d, f**). Western blot experiments were repeated three times independently, with similar results (**b**). CM, conditioned medium. EV extracellular vehicle, PP pyrvinium pamoate, HSPC hormone-sensitive prostate cancer, NEPC Neuroendocrine prostate cancer. Source data are provided as a Source Data file.

the protein bands. For Western blot, the primary antibodies used are listed in Supplementary Table 6.

## Quantitative PCR

Total RNA from cells, tissues, and extracellular vesicles were extracted with TRIzol Reagent (15596026, Invitrogen), and the concentration of RNA was quantified by a NANODROP 2000c spectrophotometer (Thermo Fisher Scientific). The reverse transcription was carried out by Reverse transcription kit (R223-1, Vazyme, China). The qPCR was performed using SYBR-green (Q711-2, Vazyme, China) in a LightCycler 480 qPCR machine (Roche). Relative gene mRNA expression level was determined using the $2^{-\Delta\Delta Cq}$ method. The primer sequences used to amplify the target genes are listed in Supplementary Data 1.

## 5-Ethynyl-2′-deoxyuridine (EdU) assay

The DNA synthesis and proliferation ability of mice tumor cells was evaluated using a BeyoClick™ EdU Cell Proliferation Kit with Alexa Fluor 594 (C0078L, Beyotime, China), following the manufacturer's instructions. We intraperitoneal (i.p.) injected EdU (50 mg/kg) 3 h before harvesting the tumors. DAPI (4′,6′-diamidino-2-phenylindole, D9542, Sigma, 1:1000) was used to stain the nuclei, and fluorescence microscope (Nikon) was used to obtain images, reflected by red and blue signals. To evaluate the proliferation ability in mice tumors, at least four nonoverlapping fields in each section were analyzed in a blinded manner. The ratio between EdU-positive and DAPI-positive cells was measured using ImageJ 1.45.

## Immunofluorescence analysis

For staining frozen samples of EdU-labeled tumors, the primary antibodies used are listed in Supplementary Table 6. In brief, the samples were fixed by 4% PFA for 15 mins, permeabilized by 0.1% Triton X-100 for 10 mins and blocked with 5% BSA/PBS at room temperature for 1 h. Primary antibodies were then incubated at 4 °C overnight, followed by rigorous wash with PBS. Double immunofluorescence staining was performed using the Alexa Fluor™ 555 Tyramide SuperBoost™ Kit (B40913, Thermo Fisher Scientific) and Alexa Fluor™ 488 Tyramide SuperBoost™ Kit (B40922, Thermo Fisher Scientific) according to the provided manufacturer's instructions. The samples were then washed rigorously with PBS. DAPI (1:1000) was added for DNA staining, and TRITC Phalloidin (40734ES80, Yeasen, China) was added for F-actin staining. Images were taken using Nikon-80i microscope under 40x objective. To quantify immunofluorescence staining, two independent researchers calculated the average number of membrane-positive cells in eight to nine random 40x fields.

## Chromatin immunoprecipitation (CHIP) assay

ChIP assay was performed using an SimpleChIP Plus Enzymatic Chromatin IP Kit (9005, Cell Signaling Technology) according to the manufacturer's instructions. Antibodies used in the ChIP assay are listed in Supplementary Table 6. Briefly, cells were crosslinked with 10% formaldehyde, then sonicated to an average length of 250–1000 bp. Then

the precipitated chromatins were decross-linked and amplified by qPCR. All the CHIP primer sets are listed in Supplementary Data 1.

## Immunohistochemistry (IHC) analysis

Prostate tumor tissues were fixed with 10% formalin for 24 hours, then embedded in paraffin and cut into 4μm series sections. Tissue sections were dewaxed and rehydrated in turn. EDTA antigen retrieval (pH 8.0) was used at 95 °C for 30 mins. Hydrogen peroxide solution (3%) was used to block endogenous peroxidase activity for 10 mins. Blocking buffer (10% horse serum in TBST) was used and incubated for 60 mins at room temperature. Slides were then incubated with specific primary antibodies diluted in blocking buffer at 4 °C overnight. Antibodies used in the IHC analysis are listed in Supplementary Table 6. The slides were then washed in TBST and incubated by corresponding secondary antibodies for 60 mins at room temperature. The sections were counterstained with hematoxylin, then dehydrated and mounted under coverslips. Representative images were taken using an Olympus light microscope.

To detect apoptotic cells, terminal deoxynucleotidyl transferase (TdT)-mediated dUTP digoxigenin nick-end labeling (TUNEL) assay was performed with TUNEL Apoptosis Assay Kits (C1098, Beyotime, China). In brief, the rehydrated sections were treated with 20 μg/mL proteinase K (37 °C, 20 mins) and then washed in 1x Tris buffer. TUNEL assay was then conducted according to the manufacturer's instructions.

For immunostaining quantification, the H-score system was obtained by multiplying staining intensity (0 for no staining, 1 for weak staining, 2 for moderate staining, and 3 for intense staining) by the percentage (0–100) of cells showing that intensity.

For antibody verification with recombinant ELAVL3 protein, anti-ELAVL3 antibody (10 ng) was incubated with purified ELAVL3 protein (100 ng) at 4 °C overnight, then followed by immunohistochemistry to validate antibody specificity.

## Cell viability assays and dose-response assays

Cell Counting Kit-8 assay was carried out to assess the viability of indicated prostate cancer cells. Six thousand cells of each treated prostate cell line were seeded into 96-well plates, and 10 μL CCK-8 reagent (A311-1, Vazyme, China) was added to each well at the appropriate time point. Then the plates were incubated in the dark incubator for 2 h. The absorbance values at 450 nm were measured with Thermo Fisher Scientific SkanIt Software 4.1. For cell viability of LASCPC-01 cells, 20000 cells were seeded in 96-well dish and treated with 100 nM pyrvinium pamoate (HY-A0293, MedChemExpress), MS-444 (HY-100685, MedChemExpress), DHTS (HY-N0360, MedChemExpress), and CMLD (HY-124828, MedChemExpress) for 72 h before performing the assay with CellTiter-Glo luminescent cell viability assay (G7570, Promega) according to the manufacturer's guidance.

For dose-response curve assays in LASCPC-01 cells, 20000 cells were seeded in 96-well dish and treated with different dosages of pyrvinium pamoate (HY-A0293, MedChemExpress) for 72 h before

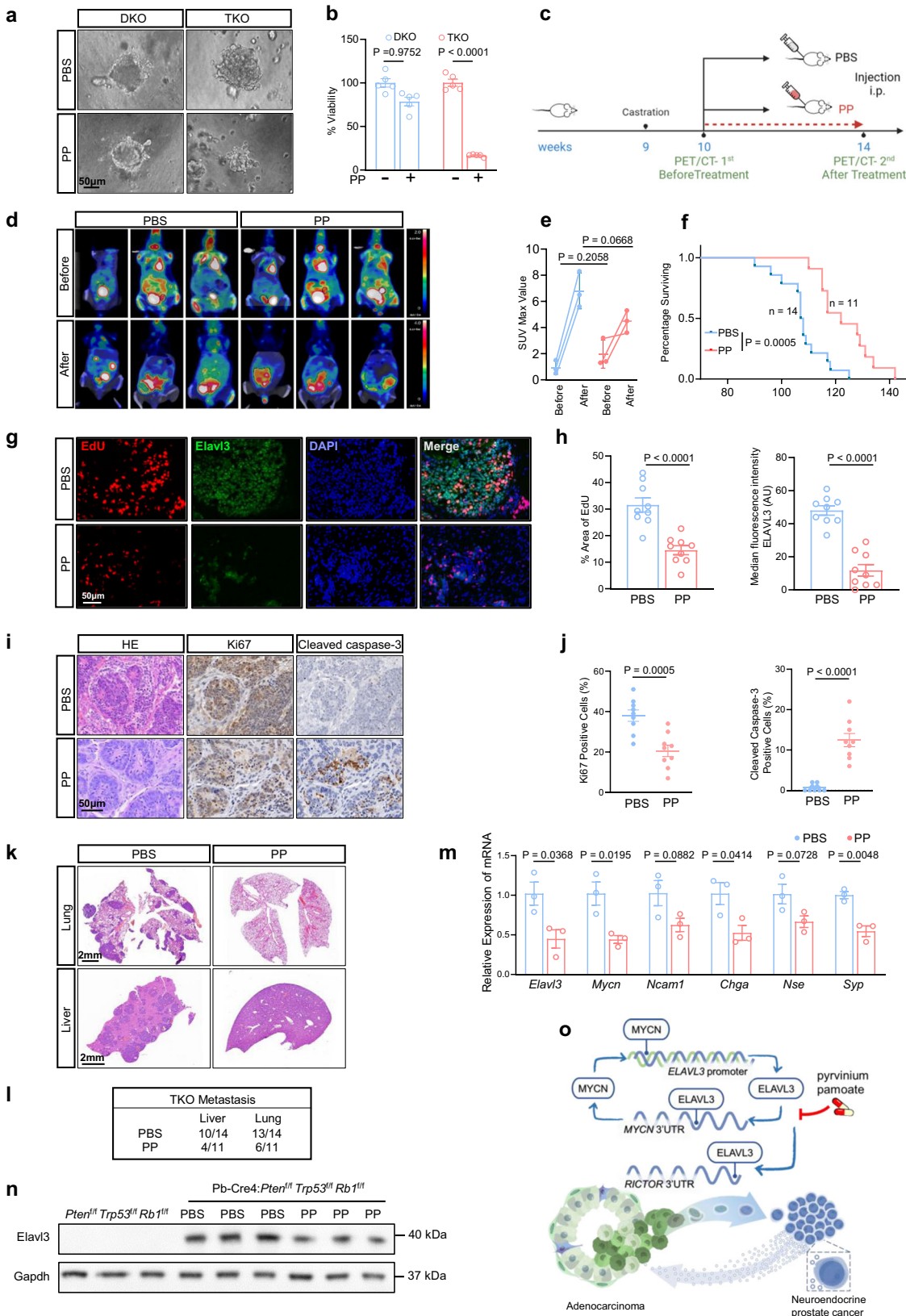

performing the assay with CellTiter-Glo luminescent cell viability assay. For dose-response curve assays in LNCaP/ELAVL3 cells and LNCaP/AR/shP53/shRB1 transduced with shRNAs cells, 8000 cells were seeded in 96-well dish and treated with different dosages of Enzalutamide (HY-70002, MedChemExpress) or MLN8237 (HY-10971, MedChemExpress) for 72 h before performing the assay with CCK-8 reagent. For dose-response curve assays in LASCPC-01/shCtrl and LASCPC-01/shELAVL3 cells, 20000 cells were seeded in 96-well dish and treated with different dosages of pyrvinium pamoate for 72 h before performing the assay with CellTiter-Glo luminescent cell viability assay. Dose-response curves were conducted and cellular 50% inhibitory concentration ($IC_{50}$) was calculated using GraphPad Prism 8.0.

**Fig. 8 | Pharmacological inhibition of ELAVL3 suppresses tumor growth and promotes survival in NEPC mouse model. a** Photograph showing morphology of organoids from Pb-Cre4: *Pten*^f/f; *Trp53*^f/f (DKO) and Pb-Cre4: *Pten*^f/f; *Trp53*^f/f; *Rb1*^f/f (TKO) prostate tumors treated with PBS or PP. Scale bar, 50 μm. **b** Cell viability of organoids from DKO and TKO prostate tumors treated with/without PP (*n* = 5 independent treated). **c** Schematic illustration showing the treatment and PET/CT imaging of TKO. **d** [^18F]-FDG PET/CT imaging of TKO before and after indicated treatment. **e** Quantification of [^18F]-FDG PET/CT signal by maximum standardized uptake value in TKO before and after indicated treatment (*n* = 3 per group). **f** Kaplan–Meier survival analysis of TKO treated with PP or PBS (PBS, *n* = 14; PP, *n* = 11). **g, h** Representative immunofluorescence staining and quantification of EdU and Elavl3 in tumors from TKO treated with PP or PBS. Scale bar, 50 μm. (PBS, *n* = 9; PP, *n* = 8). **i, j** Representative H&E, immunohistochemistry staining and quantification of Ki67 and cleaved-caspase 3 in tumors from TKO treated with PP or PBS. Scale bar, 50 μm. (*n* = 9 per group). **k** Representative H&E staining of lung and liver metastases from TKO treated with PP or PBS (*n* = 3 per group). Scale bar, 2 mm. **l** Summary of the liver and lung metastases from TKO treated with PP or PBS. **m, n** QPCR showing relative mRNA expression of indicated genes and western blot showing indicated protein expression in tumor tissues from TKO treated with PP or PBS (*n* = 3 per group). **o** Schematic showing that ELAVL3 is transcriptionally activated by MYCN, and then in turn binds to and stabilizes the mRNA of *MYCN* and *RICTOR*. ELAVL3 can be released into EVs and induce neuroendocrine differentiation of recipient cells. PP disrupts the interaction of ELAVL3 and MYCN, reduces ELAVL3-induced neuroendocrine differentiation, and provides a promising drug repurposing strategy for the treatment of NEPC. Data presented as mean ± s.e.m. (**b, e, h, j, m**). Statistical significance was determined by two-tailed unpaired Student's *t*-test (**b, e, h, j, m**), or log-rank (Mantel-Cox) test (**f**). Western blot experiments were repeated three times independently, with similar results (**n**). Source data are provided as a Source Data file.

## Plasmids and lentiviral production

The plvx304-ELAVL3-V5-blasticidin-CMV, plvx304-MYCN-HA-puromycin-CMV, and pBabe-AR-puro-CMV were purchased from the Core Facility of Basic Medical Sciences, Shanghai Jiao Tong University School of Medicine, with empty vector as control. RRM1, RRM2, RRM3 and hinge region deletion of ELAVL3 were obtained from PCR-directed mutagenesis using KOD-plus Kit (TOYOBO) using the wildtype plasmid as the template. The plvx304-ELAVL3-GFP-blasticidin-CMV expression plasmid was constructed by inserting the GFP sequence into the plasmid plvx304-ELAVL3-V5-blasticidin-CMV.

Human ELAVL3, MYCN, RICTOR, TP53, and RB1 short-hairpin RNA (shRNAs) were cloned into PLKO.1 plasmid, respectively. The sequences used are listed in Supplementary Data 1.

All plasmids were verified by DNA Sanger sequencing. Lentivirus was prepared using a three-plasmid packing system. Briefly, PLKO.1 or plvx304 vectors were co-transfected into HEK293T cells along with expression vectors containing the psPAX2 and pMD2G genes.

Lentivirus was harvested at 48 and 72 h after transfection, and the virus was cleansed by the 0.45-μm filter. Stable cell lines were selected out in 1–2 μg/mL puromycin or 10 μg/mL blasticidin S for 1 week.

To generate single-cell clones with endogenous ELAVL3 promoter-deficient cells, a CRISPR/Cas9 approach was conducted. A Cas9-stable HEK293T/sgCtrl cells were generated with lenticrispr-v2 (83480, Addgene). Then, we designed two guide RNAs (listed in Supplementary Data 1) targeting 80 bp sequence in the promoter region of ELAVL3, and transfected them into HEK293T/sgCtrl cells. Single-cell clone selection was initiated 48 h after transduction. Cells transfected with control sequence were treated in a similar manner to obtain ELAVL3-promoter wildtype single-cell clones.

## RNA Pull-down assay

Based on luciferase reporter results, we designed six fragments RNA probes of *MYCN* mRNA and *RICTOR* mRNA, respectively. The biotin-coupled probe RNA pull-down assay was performed with Magnetic RNA-Protein Pull-Down Kit (20164, Thermo Fisher Scientific) according to the manufacturer's protocol. In brief, a total of $2 \times 10^7$ LNCaP/ELAVL3, HEK293T/ELAVL3, or PC3/ELAVL3 cells were harvested, lysed and incubated with 50 pmol biotin-coupled probes for each, which were pre-bound on magnetic beads at 4 °C for one night. After washing with wash buffer, the bead-binding proteins were eluted by boiling in the loading buffer and subjected to SDS-PAGE. The ELAVL3-V5 and LaminB1 proteins were detected with Western blot. The biotin-coupled RNA probes for *MYCN* mRNA 3'-UTR and *RICTOR* mRNA 3'-UTR were purchased from Sangon. Sequences are listed in Supplementary Data 1.

## RNA electrophoretic mobility-shift assay (RNA-EMSA)

To generate recombinant ELAVL3 protein, ELAVL3 was amplified and subcloned into the pGEX-4T-1 vector. E. Coli BL21 cells were transformed with pGEX-4T-1 plasmid and subsequently induced with 0.5 mM isopropyl-β-D-thiogalactoside (I5502, Sigma) at 22 °C for 4 h. Following induction, cells were harvested and resuspended in cold PBS supplemented with 1 mg/mL lysozyme (L1667, Sigma) and protease inhibitor cocktail (11836145001, Roche), followed by sonication (16 cycles of 15 s each at 30% output). The supernatant was separated using glutathione-sepharose beads (G0924, Sigma-Aldrich) according to the manufacturer's instructions. The RNA-EMSA was performed by following the manufacturer's instruction of RNA EMSA kit (GS606, Beyotime, China). Briefly, recombinant ELAVL3 protein and 3'biotin-coupled RNA were incubated at 37 °C for 1 h. Reactions were carried out in a 10 μL buffer containing 20 mM Tris-HCl, 100 mM NaCl, 5 mM MgCl₂, 2 mM DTT. The bound complexes were loaded onto a 6% non-denaturing polyacrylamide gel at 4 °C, and transferred onto a nylon membrane. Subsequently, the membrane was subjected to UV irradiation at a dose of 120 mJ/cm², and northern blot analysis was conducted to detect the target RNA. The biotin-coupled RNA probes and non-labeled RNA probes were purchased from Sangon, sequences are listed in Supplementary Data 1.

## mRNA decay assay and nascent RNA pulse-labeling

For the mRNA decay assay, cells (PC3/Ctrl, PC3/MYCN, PC3/MYCN-3'-UTR, LASCPC-01/Ctrl, LASCPC-01/ELAVL3, LASCPC-01/shCtrl, LASCPC-01/shELAVL3) were seeded into 6-well plate ($2 \times 10^5$ cells per well). After transfection with ELAVL3 or Ctrl plasmid for 48 h, Actinomycin D (5 μg/mL, A4448, Apexbio) was added to each well and incubated for indicated time. Total RNA was then extracted using TRIzol reagent following the manufacturer's instructions and the remaining transcript was determined by qPCR.

For nascent RNA pulse-labeling, the indicated cells were transfected with ELAVL3 or Ctrl plasmid for 48 h and then chased by 200 μM 4sU (E1292, Selleck) in growth medium to label nascent RNA. Two hours after pulsing, cells were washed with PBS three times and fresh growth medium was added. Cells were harvested at indicated time, and total RNA was extracted using TRIzol reagent and quantitated. For each sample, 100–200 ug of RNA was biotinylated using EZ-link Biotin-HPDP (21341, Thermo Fisher Scientific) in 0.2 mg/mL DMSO and biotin labeling buffer (10 mM Tris pH 7.4, 1 mM EDTA), incubating for 1.5 hours at RT under rotation. The biotin-labeled RNA was enriched by chloroform-isoamyl alcohol extraction, resuspended in DEPC-treated ddH₂O, and pulled down using Dynabeads MyOne Streptavidin C1 (65001, Thermo Fisher Scientific) in Dynabeads binding and washing buffer (1 M NaCl, 5 mM Tris pH 7.5, 0.5 mM EDTA) for 5 mins and 6 times at RT with rotation. After stringent washing, the 4sU-labeled RNA was eluted from the beads with freshly prepared 100 mM DTT in DEPC-treated ddH₂O and purified using the RNeasy MinElute kit (74204, QIAGEN). The cDNA was synthesized from the 4sU-labeled RNA using SuperScript IV VILO Master Mix (11756050, Thermo Fisher Scientific) and analyzed by qPCR.

## Dual-luciferase reporter assay

For the construction of pMIR-MYCN-3′-UTR, pMIR-RICTOR-3′-UTR, and their indicated segments, PCR products were prepared with primer (listed in Supplementary Data 1) targeting indicated 3′-UTRs fragments of *MYCN* or *RICTOR* mRNA and cloned into the NheI/SalI site of plasmid pMIR-GLO reporter vector (H306, Obio Technology, China).

For the construction of pGl4.10-ELAVL3, BS1, and BS2 mutation plasmid, PCR products were prepared with primers (listed in Supplementary Data 1) targeting indicated promoter sequence of ELAVL3 and cloned into the XhoI/HindIII site of pGl4.10 reporter vector (E6651, Promega).

HEK293T cells were seeded onto 24-well plates ($1 \times 10^5$ cells per well) and allowed to grow overnight. Then the transient transfection of indicated plasmids containing Renilla (E1910, Promega) was carried out using Lipofectamine 3000 (Invitrogen). After 48 h, luciferase activity was conducted with the Dual Reporter Luciferase Assay System (E1910, Promega), according to the manufacturer's instruction. The relative levels of luciferase activity were normalized to the levels of Renilla luciferase activity and the control group 48 h after transfection.

## RNA-sequencing

Cells (LNCaP/AR/shP53/shRB1/shCtrl versus LNCaP/AR/shP53/shRB1/shELAVL3 and NCI-H660 treated with pyrvinium pamoate versus treated with PBS) were subjected to HiSeq RNA-seq (Cloud-seq Biotech Ltd., China) with two duplicates of each group. Briefly, total RNA was extracted using TRIzol reagent according to the manufacturer's instructions. After treating with DNase I, mRNA was enriched by Oligo(dT) modified magnetic beads and served as templates for cDNA synthesis (Invitrogen). RNA integrity was measured and the library was quantified using Agilent 2100 Bioanalyzer (Agilent, Santa Clara, CA). Sequencing was performed on the Illumina HiSeq2000 system (Illumina, San Diego, CA) at a high output model according to the manufacturer's instruction. Raw data were filtered by removing reads of low quality (over 10% uncertain bases or over 50% bases of Q < 5) or containing adapters, and approximately 20 million clean reads per sample were sampled for following analysis. Transcriptome reads were mapped to the reference genome (hg19) using the Bowtie tool. The gene expression level was quantified using RSEM software. The significance of differentiated expression was determined by a P-value threshold of 0.05. FPKM of each gene was calculated using Cufflinks, and the read counts of each gene were obtained by HTSeq-count.

## Bulk transcriptome analysis

Differentially expressed genes between the two groups were calculated in R v4.0.5 by limma package v3.46.0. *P* < 0.001 with log2FoldChange >1 or <−1 was set as cutoff. Gene set enrichment analysis was performed on GSEA function of clusterProfiler package v3.18.1 or GSEA software v4.1.0 to analyze the biological difference. The enrolled gene sets included Gene Ontology, Hallmark, and Kyoto Encyclopedia of Genes and Genomes adopted from the Molecular Signatures Database.

## Single-cell RNA sequence reanalysis

The raw gene expression matrix of single-cell transcriptome was derived from GSE137829 and further analyzed in R 4.0.5 using Seurat package v4.0.2. The data from 6 castration-resistant prostate cancer samples were integrated using the canonical correlation analysis approach. Cells with all the following features were filtered after Seurat-based quality control: (1) 500–7,000 detected genes; (2) < 10% mitochondrial gene expression; (3) < 3% red blood cell gene expression. Epithelial cells were identified and extracted based on the expression level of *EPCAM*, *KRT5*, *KRT8*, *KRT14*, *KRT18*, and *CDH1*. Unsupervised cell clusters were identified by shared nearest neighbor modularity optimization-based clustering algorithm. Differentially expressed genes between the high-neuroendocrine clusters and the low-neuroendocrine clusters were calculated by using the FindMarkers function in the Seurat package. *P* < 0.001 with log2FoldChange >1 or <−1 was set as the cutoff.

## Ribonucleoprotein immunoprecipitation (RIP), sequencing and analysis

To perform RNA-binding protein ELAVL3 immunoprecipitation assay, PC3, DU145, and HEK293T cells were cultured in five 10-cm dishes and transfected with plvx304-CMV-ELAVL3-V5 or control expression vectors. After 48 h, about $1 \times 10^7$ cells in each dish were harvested and resuspended in 250 μL lysis buffer [20 mM Tris (pH 8.0), 150 mM NaCl, 1% NP-40, 10% glycerol, 2 mM EDTA, Protease inhibitor cocktail (11836145001, Roche), 0.5 mM PMSF (ST506, Beyotime, China), 2 mM Ribonucleoside Vanadyl Complexes (R0108, Beyotime, China), 2 μL RNase inhibitor (R0102, Beyotime, China)]. 10% volume of the lysates was separated into another two RNase-free tubes, labeled as input, and the rest lysates were used for RIP assay. All lysates were frozen rapidly at −80 °C overnight to lyse the cells gently. Then the RIP lysates were thawed quickly and centrifuged at 15,000 rpm for 10 mins at 4 °C. The supernatant was collected and incubated with V5 antibody (13202, Cell Signaling Technology) and protein A + G magnetic beads (HY-K0202, MedChemExpress) at 4 °C for 4 h with rotation. The V5-Beads were washed with 500 μL lysis buffer for 5 times rigorously. After the last washing, thawed the input lysates and added 1 mL TRIzol reagent to the input lysates and the V5-Beads to extract the total RNA. RNA was extracted using TRIzol reagent according to the manufacturer's instructions. rRNAs were removed by using Ribo-Zero™ rRNA Removal Kit (Illumina, San Diego, CA, USA). RNA library was constructed using rRNA-depleted RNAs with the TruSeq Stranded Total RNA Library Prep Kit (Illumina, San Diego, CA, USA) according to the manufacturer's instructions. RNA integrity was measured and the library was quantified using Agilent 2100 Bioanalyzer (Agilent, Santa Clara, CA). Sequencing was performed on the Illumina HiSeq2000 system (Illumina, San Diego, CA) at high output model according to the manufacturer's instruction. Raw reads were filtered with Fastp (v0.20.0), an ultra-fast all-in-one FASTQ preprocessor, to obtain high-quality clean reads by removing sequencing adapters, short reads (length <30 bp) and low-quality reads. Then FastQC was employed to ensure high reads quality and to process data quality control. Differentially expressed genes between the two groups were calculated in R v4.0.5 by limma package v3.46.0. *P* < 0.05 with log2FoldChange >1 or <−1 was set as cutoff. Gene set enrichment analysis of the differentially expressed genes was performed on the web-based Metascape[25] via www.metascape.org.

## Tumor xenograft model

All male BALB/c nude mice were provided by the animal laboratory of Ren Ji Hospital and maintained in a specific pathogen-free facility licensed by Shanghai Science and Technology Commission (SYXK[hu] 2016-0009), and were randomly separated into two or three groups. Every precaution has been taken to minimize animal distress. The allowed maximal tumor volume is 2000 mm³ in accordance with institutional tumor production policies, and was not exceeded in all our experiments.

To evaluate the influence of ELAVL3, $5 \times 10^6$ LASCPC-01/shCtrl and LASCPC-01/shELAVL3 cells were suspended in DMEM/Matrigel (1:1; Corning) and implanted into the right armpit of 4 weeks old male BALB/c nude mice (100 μL per mice). Three weeks after inoculation, the tumor volume (calculated as Volume = $0.52 \times$ Length $\times$ Width²) of the prostate tumor xenografts was measured using an electronic vernier caliper every 3 days. The mice were sacrificed at week 8 and tumors were collected and processed for downstream analyses by overnight fixation with 10% buffered formalin followed by tissue processing and embedding in paraffin.

To evaluate the influence of pyrvinium pamoate, $2 \times 10^6$ DU145/ELAVL3 cells were suspended in DMEM/Matrigel (1:1; Corning) and implanted into the right flank of 4 weeks old male BALB/c nude mice (100 μL per mice). On week 3, mice were randomly separated into 2 groups, followed by i.p. administration PBS (4 times per week for 5 consecutive weeks, $n = 5$) or pyrvinium pamoate (P0027, Sigma Aldrich, 0.2 mg/kg per injection, 4 times per week for 5 consecutive weeks, $n = 5$). The tumor volume of the prostate tumor xenografts was measured using electronic vernier caliper every 3 days. The mice were sacrificed at week 8 and tumors were harvested carefully, photographed, and processed for downstream analyses by overnight fixation with 10% buffered formalin followed by tissue processing and embedding in paraffin.

## Animals

All animal experiments in the current study were performed according to the ethical regulations of Ren Ji Hospital and maintained in a specific pathogen-free animal facility. Room temperature was maintained at 21–22 °C and the humidity at 25–47%. Animals were kept at a 12 h/12 h dark/light cycle and maintained in ventilated cages (4 mice per cage). Animal experiment protocols were approved by the Ren Ji Hospital Laboratory Animal Use and Care Committee. Tg(Pbsn-cre)4Prb/J (026662), B6.129P2-*Trp53*[tm1Brn]/J (008462), *Rb1*[tm2Brn]/J (026563), and B6.129S4-*Pten*[tm1Hwu]/J mice (006440) were bought from Jackson Laboratory to generate Pb-Cre4: *Pten*[f/f]; *Trp53*[f/f]; *Rb1*[f/f] mice, Pb-Cre4: *Pten*[f/f]; *Trp53*[f/f] mice, and *Pten*[f/f]; *Trp53*[f/f]; *Rb1*[f/f] mice. All strains were on the C57BL/6 background. Only male mice were utilized in the study, and their sex was determined based on similar studies in prostate cancer. PCR Primers for Pb-Cre4: *Pten*[f/f]; *Trp53*[f/f]; *Rb1*[f/f] mouse genotyping are listed in Supplementary Data 1. After 8 weeks, castration was initiated with anesthetized with 1% isofluorane. Surgery was performed on a heating pad until mice completely recovered from anesthesia. To test the therapeutic effect of pyrvinium pamoate on Pb-Cre4: *Pten*[f/f]; *Trp53*[f/f]; *Rb1*[f/f] mice, pyrvinium pamoate was i.p. administrated to castrated mice (0.2 mg/kg per injection, 4 times per week) for 4 weeks. All mice were sacrificed at approximately 8 weeks after castration. Then the prostate, liver, and lung were harvested carefully, photographed, and processed for further experiments.

The Luc.Cre empty vector (20905, Addgene), and lentivirus was packaged on HEK293T cells along with expression vectors containing the psPAX2 (12260, Addgene) and pMD2G (12259, Addgene) genes. Viral supernatant was concentrated by ultracentrifugation (2 h at $20,000 \times g$), then an in vitro infection test was conducted in advance. An anesthetized and surgically prepared animal was placed in dorsal recumbency. When *Pten*[f/f]; *Trp53*[f/f]; *Rb1*[f/f] mice reached 8 weeks old, intra-prostate infection was initiated. After exposure to 1% isoflurane (R510-22, RWD, China) for anesthesia, the lower half of the abdomen was shaved and the mouse was placed in a surgery hood. The shaved region was cleaned with betadine and a 2 cm incision in both the skin and peritoneum was made along the lower abdominal midline to allow the right anterior prostate to be positioned for injection on a sterile support. Typically, 25 μL of concentrated Luc.Cre lentivirus was injected into the right anterior prostate. The incision was then sutured and the skin was stapled shut using 2 stainless steel EZ Clip wound closures. Surgery was performed on a heating pad until mice completely recovered from anesthesia.

## [$^{18}$F]-FDG PET/CT and μCT

One week after castration surgery, mice were subjected to PET/CT analysis. PET/CT imaging was performed on an Inveon MM Platform (Siemens Preclinical Solutions, Knoxville, Tennessee, USA) with a computer-controlled bed and 8.5 cm trans-axial and 5.7 cm axial fields of view (FOV). The animals were anesthetized with 1% isoflurane in $O_2$ gas for [$^{18}$F]-FDG injection. A single injection contains 0.1 mL FDG with

an activity of 10 MBq intravenously in the tail vein, immediately awakened afterward and placed back in the anesthesia cage. One hour after administration of the tracer injection, animals were anesthetized with isoflurane, placed prone on the PET scanner bed near the central field of view, and were maintained under continuous anesthesia during the study with 1% isoflurane in $O_2$ gas at 2 L/min. Inveon Acquisition Workplace (IAW) 1.5.0.28 was used for scanning process. Mice were scanned 10 mins CT X-ray for attenuation correction with a power of 80 Kv and 500 uA and an exposure time of 1100 ms before PET scan. Ten-minute static PET scans were then acquired, and images were reconstructed by an OSEM3D (Three-Dimensional Ordered Subsets Expectation Maximum) algorithm followed by MAP (Maximization/Maximum a Posteriori) or FastMAP provided by IAW. The 3D regions of interest (ROIs) were drawn over the heart guided by CT images and tracer uptake was measured using the software of Inveon Research Workplace (IRW) 3.0. Individual quantification of the [$^{18}$F]-FDG uptake in each was calculated. Max standardized uptake values (SUV) were determined by dividing the relevant ROI concentration by the ratio of the injected activity to the body weight. Then the mice were randomly separated into two groups followed by i.p. administrated PBS (4 times per week for 4 consecutive weeks, $n = 3$) or pyrvinium pamoate (0.2 mg/kg per injection, 4 times per week for 4 consecutive weeks, $n = 3$). After 4 weeks of treatment, mice were subjected to PET/CT analysis again for paired pre- and post-treatment tumor volume.

## Bioluminescence Imaging

In vivo bioluminescence imaging was performed using a Xenogen IVIS Spectrum imager, which utilizes a highly sensitive, cooled CCD camera mounted in a light-tight camera box. Mice received luciferin (40901, Yeasen, China) at 200 mg/kg by i.p. 10 mins prior to imaging, then anesthetized using 1% isoflurane and placed onto the warmed stage inside the camera box, followed by 1 min of imaging. For quantification, regions of interest (ROI) were measured with standardized rectangular regions covering the mouse abdomen. The measured signal was quantified as photons/second (ph/sec) using the Living Image software v.4.2 (Xenogen). Background bioluminescence in vivo was in the range of $2–5 \times 10^4$ ph/sec. Then the mice were randomly separated into two groups according to the ROI values, followed by i.p. PBS (4 times per week for 8 consecutive weeks, $n = 4$) or pyrvinium pamoate (0.2 mg/kg per injection, 4 times per week for 8 consecutive weeks, $n = 4$). After 8 weeks treatment, mice were subjected to bioluminescence imaging again to acquire the pre-treatment and post-treatment tumor imaging and ROI values.

## Extracellular vesicles isolation from cells medium and patient serum

Cells were cultured in RPMI 1640 media or MEM media, each supplemented with 1% pen-strep and 10% of exosome depleted FBS (#A27208-01, Gibco) at 37 °C, 5% $CO_2$ for 48 hours before collecting the conditioned medium for exosome isolation. Cells were cultured in 10 cm dishes in 10 mL of media. At the time of collection, cells were 80–90% confluent. All the further steps for exosome isolation were carried out at 4 °C. The conditioned medium was centrifuged at $1000 \times g$ for 10 mins (5810 R, Eppendorf) and then filtered using a 0.22 μm syringe-filter (SLGP033RS, Millipore) to eliminate cells and large debris. Next, the supernatant was centrifuged at 10,000 x g for 60 mins at 4 °C to remove small debris and other large EVs. Third, the supernatant was transferred to ultra-clear ultracentrifuge tubes (344058, Beckman Coulter) and then ultracentrifuged at $120,000 \times g$ for 2 h at 4 °C (Optima XE-90 Ultracentrifuge, Beckman Coulter), followed by washing with PBS to resuspend the EV pallet and another ultracentrifugation at $120,000 \times g$ for 2 h at 4 °C in the same tube. After the removal of supernatant, the EV pellets were resuspended and collected in 100 μL PBS and stored at −80 °C until further processing

for downstream application. To remove non-vesicular proteins, EVs were treated with 100 µg/mL proteinase K (ST533, Beyotime, China) for 1 hour followed by heat inactivation of the protease.

For density gradient isolation, EV pellets were resuspended in 1.5 mL of suspension buffer (0.25 M sucrose, 10 mM Tris pH 8.0, 1 mM EDTA pH 7.4) and mixed in a 1:1 ratio with a 60% stock solution of iodixanol/Optiprep (D1556, Sigma Aldrich). Stepwise layering of different iodixanol concentrations was then performed as follow: 1.4 mL of 40% iodixanol, 1.3 mL of 20% iodixanol, and 1.2 mL of 10% iodixanol were gently layered on top of the EV suspension. Subsequently, the tubes were ultracentrifuged at 120,000 × g overnight at 4 °C. Then, ten fractions of 700 µL each were collected from the top of the tube individually, washed with PBS, and subjected to ultracentrifugation at 120,000 × g for 2 h at 4 °C. Finally, the pellets were resuspended in 40 µL lysis buffer for immunoblotting.

### Transmission electron microscopy (TEM)

TEM was used to detect the morphology of separated EVs. Briefly, mixed 50 µL of EVs sample with an equal amount of 4% paraformaldehyde to complete fixation. A 5 µL drop of EVs was adsorbed to a 200-mesh copper mesh and incubated for 5 min at room temperature in a dry environment. The excess liquid was then removed from the edge of the copper mesh by aspiration with filter paper. Next, the sample was stained with 1% uranyl acetate with methylcellulose for 1 min. The copper mesh was rinsed with distilled water twice, and excess liquid was removed with filter paper. After drying, the copper mesh was observed with transmission electron microscope (HT7800, HITACHI) at 80 kV.

### Nanoflow cytometry analysis

Size distribution and concentration of EVs samples were analyzed by Nano Flow Cytometry according to the reported protocol[53]. According to the manufacturer's standard process, the instrument used Silica Nanosphere Cocktail (NanoFCM, Inc., S16M-Exo) to calibrate for size distribution while using 250 nm PE and AF488 fluorophore-conjugated polystyrene beads for granule concentration. All EVs samples were diluted to make sure the number of granules was within the optimal range of 4000–8000/min. Then we recorded the number of particles passing through the instrument in one minute for each sample. Finally, relevant size and concentration were acquired after the conversion of the flow rate and side-scattering intensity of granules based on calibration curve on the NanoFCM software (NanoFCM Profession V2.0).

### Statistical analysis and reproducibility

All statistical analyses were performed with GraphPad 8.0 software. Two-tailed Student's $t$-test assuming equal variance was used, and one-way analysis of variance for independent variance. $P < 0.05$ was considered significant. Data are presented as means ± s.e.m. All animals were randomized and exposed to the same environment. Blinding was not performed in tumor measurement and IHC staining. Western blot experiments were repeated at least three times, and representative images were shown.

### Reporting summary

Further information on research design is available in the Nature Portfolio Reporting Summary linked to this article.

## Data availability

The RIP-seq and RNA-seq data generated in this study have been deposited in the NCBI database under accession code GSE224911. The publicly available prostate cancer clinical data and RNA-seq data used in this study are available in the cBioPortal database[23] (www.cbioportal.org) and GEO database under accession code: GSE90891[22], GSE86532[4], and GSE137829[21]. The publicly available ChIP-seq data used in this study are available in the GEO database under accession code:

GSE117306[28]. The remaining data are available within the Article, Supplementary Information or Source Data file. Source data are provided with this paper.

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

## Acknowledgements

We are grateful to the patients for making available the tumor samples that contribute to this research. We thank Dr. Jinke Cheng, Dr. Jianxiu Yu, and Dr. Xian Zhao for their suggestions. We thank Lucky Wong for discussion of this study. We thank the Core Facility of Basic Medical Sciences, Shanghai Jiao Tong University School of Medicine for the plasmids. We thank Cloud-seq Biotech Ltd. Co. (Shanghai, China) for the mRNA-seq service and the subsequent bioinformatics analysis. The work was supported by National Natural Science Foundation of China (82172868 to Q.W., 81972578 to Q.W., No 82072847 to W.X.), the Program for Professor of Special Appointment (Eastern Scholar) at Shanghai Institutions of Higher Learning (TP2017029 to Q.W.), Shanghai Municipal Education Commission Grant Support (20191906 to W.X., 20171912 to Q.W., 2023ZKZD23 to W.X.), Science and Technology Commission of Shanghai Municipality (19XD1402300 to W.X.), Shanghai Municipal Health Commission (2019LJ11 to W.X., 2020CXJQ03 to W.X.), Ren Ji Hospital (LYZXHXKT220845 to W.X., 2020LYRJ-002 to W.X., PNO-0106 to W.X., RJKY18-02 to W.X.).

## Author contributions

Q.W. and W.X. conceived and designed the study. Y.J. and K.S. collected and assembled data. W.Z., Y.J., B.L., and Z.M. collected mouse data. L.D., Y.Z., H.H.Z., B.D., J.P. and Z.J. provided study materials or patient samples and Y.Y., A.L., and Z.X. analyzed and interpreted the data. R.S. and Z.M. analyzed single-cell data. Y.J. and R.S. processed and analyzed RIP-seq data and RNA-seq data. K.S., Y.X., X.L., and C.H. participated in immunostaining and imaging. W.X. and Q.W. supported data analysis. W.X., Q.W., and Y.J. interpreted the data and wrote the manuscript.

## Competing interests

The authors declare no competing interests.
