## [Peer Review File · Nature Communications]

Reviewers' Comments:

Reviewer #1:

Remarks to the Author:

By performing single cell RNA-seq in six CRPC patients including two NEPC tumors, the authors identified ELAVL3 as one of the NEPC associated genes. They further confirmed this observation in different published NEPC cohort. Mechanistically, the authors showed that expression of ELAVL3 is transcriptionally regulated MYCN. Intriguingly, they further showed that ELAVL3 binds and promotes stability of MYCN mRNA. Finally, the authors showed that pharmacological targeting of ELAVL3 not only inhibited NEPC phenotype, but also inhibited growth of NEPC cells in vitro and in vivo.

Overall, the findings are novel and clinically relevant. Also, the conclusion is generally supported by the data provided. However, a few issues need to be addressed to further improve the study.

Major points:

ELAVL3 RIP-seq experiments were performed in PC-3 and DU145 cells with or without ectopic expression of ELAVL3. Why did the authors perform such experiments in H660 cells which express higher level of endogenous ELAVL3? At least some of the targets of ELAVL3 be verified by RIP-qPCR in H660 cells.

By performing RIP-qPCR analysis, the authors showed that ELAVL3 binds to MYCN mRNA. Can this observation be confirmed by the ELAVL3 RIP-seq data?

Explanation should be provided as to why different cell lines were used in different experiments such as RIP-seq, RIP-qPCR, ELAVL3 binding of MYCN mRNA and ELAVL3 regulation of MYCN mRNA stability.

The color differences for different RRM domain deletion mutations in Figure 4h are too subtle to be distinguished.

Is RICTOR mRNA a binding target of ELAVL3 which can be revealed by ELAVL3 RIP-seq studies?

It is unclear that the anti-cancer effect of pyrvinium pamoate (PP) is mediated through ELAVL3.

Minor points:

English and grammar need to be polished and/or edited.

Reviewer #2:

Remarks to the Author:

This paper is focused on a novel pathway mediated by ELAVL3 and MYCN that is claimed to promote neuroendocrine prostate cancer.

The Authors claim that that the RNA-binding protein ELAVL3 is specifically upregulated in NEPC, and that ELAVL3 induces a neuroendocrine phenotype in prostate adenocarcinoma; that it binds to and stabilizes the mRNA of MYCN and RICTOR; that ELAVL3 can be released in extracellular vesicles. They finally claim that ELAVL3 promotes tumor growth.

Although many data are provided, the results are often presented as graphs and not as original data. Furthermore, major weaknesses are highlighted by this Reviewer. The Authors' conclusions are therefore not supported by the data shown.

MAJOR POINTS

- 1) The immunohistochemical analysis is performed with an antibody that needs to be validated by competition with the antigen/antigen peptide detected
- 2) Fig 1f indicates that the H660 cell line expresses high levels of ELAVL3; some of the other cell lines express a lower MW band. How do the Authors explain this discrepancy?
- 3) Fig 2 shows only 1shRNA tested in vivo

- 4) The weak specificity of the drug used, PP, is a concern
- 5) The Extended 5b image is not optimal and the data are not quantified
- 6) The data in the Extended 6d need to be supported by an immunoblot
- 7) Fig 6d: protease treatment is needed to confirm that the protein is inside the extracellular vesicles
- 8) Fig 6: Further isolation of extracellular vesicles by density gradient should be used to confirm association of ELAVL3 with extracellular vesicles
- 9) TKO mice are not characterized
- 10) The Extended 6e data are not described; is ELAVL3 absent in those samples?

Reviewer #3:

Remarks to the Author:

Review

In this manuscript, Ji et al., report studies arguing that the RNA-binding protein ELAV3 and the MYCN oncoprotein participate in a feed-forward loop that drives neuroendocrine differentiation of prostate cancer cells—resulting in development of deadly neuroendocrine prostate cancer (NEPC). They argue that ELAV3 is overexpressed in NEPC, that its overexpression drives the NEPC phenotype, that it is a transcriptional target of MYCN, that it stabilizes the MYCN mRNA, and that its functions in vitro and in mice can be blocked by the FDA-approved parasitic worm medication pyriminium pamoate.

The topic is interesting and potentially important. New treatment options are desperately needed for NEPC, and repurposing pyriminium pamoate for this purpose is an exciting option (not novel in the cancer realm in general, but for this cancer type it is novel). The connections between MYCN and ELAV3 are also potentially exciting and interesting.

That said, multiple major technical concerns undermine all of the major conclusions of the work.

Major concerns:

Figure 1F. The authors claim high basal level expression of ELAV3 in LASCPC-01 and H660 cells at the protein level. But there is no apparent expression of ELAV3 in any of the lines except H660. As LASCPC-01 cells are NEPC-derived, the absence of ELAV3 expression in this setting challenges the generality of the results presented in the manuscript. This issue is further problematic due to the fact that H660 cells express high levels of ELAV3, but have no detectable MYCN protein expression.

Figure 2D. shRNA knockdown of ELAV3 in H660 cells needs to be done with a second shRNA (as in LASCPC-01 cells). A western blot needs to also be performed to show knockdown at the protein level.

Figure 2E. The western blot on LASCPC-01 cells in this part of the figure appears at odds with that shown in Fig. 1F. In the latter, we see high levels of MYCN and no expression of ELAV3. Here, we see expression of ELAV3 but MYCN appears barely above background. What is the cause of this discrepancy?

Figure 2J and L. The data connecting ELAV3 expression to enzalutamide are weak. The effects of ELAV3 knockdown on LAPR viability and response to enzalutamide are subtle, and the statement that ELAV3 "facilitates the acquisition of enzalutamide resistance" is made without sufficient justification. The conclusions need to be toned down.

Figure 3. One of the key conclusions in the manuscript is that ELAV3 is a direct transcriptional target of MYCN. Indeed, this is the bedrock of the story. Yet the data supporting this conclusion are weak, and based largely on (i) transient reporter assays that do not necessarily reflect faithfully reflect events in endogenous genes, (ii) long term knockdown experiments that conflate primary, secondary, and tertiary effects, and (iii) indirect measures of transcriptional output. The flimsy nature of these data undermines the major conclusions of the work. To support this conclusion, candidate E-box elements need to be mutated in the endogenous ELAV3 promoter,

acute perturbations of MYCN need to be used, and some measure of transcription (not transcripts, not reporter proteins) needs to be performed. The same general criticism applies to the indirect/artificial assays used to probe regulation of MYCN mRNA in Figure 4.

Figure 4A. Western blots need to be included to show knock-down of MYCN at the protein level. We also need to see how this impacts ELAV3 protein expression.

Figure 4F. The authors conclude that ELAV3 directly binds to the MYCN mRNA. This, again, is a foundation of the story in the manuscript. Yet there are no direct ELAV3–mRNA binding assays here; just a pulldown from crude lysates. The authors need to show a direct interaction between recombinant ELAV3 and purified mRNA to support this important claim.

Figure 4G. An actinomycin D assay is used to infer that ELAV3 regulates the stability of MYCN mRNA. These assays are prone to artifacts associated with the use of a potent transcriptional poison. The effect on MYCN mRNA stability needs to be shown with a second, more focused (non-poison) approach such as pulse-labeling or a specific promoter shutoff.

This second of the manuscript concludes with the statement that ties regulation of MYCN mRNA stability directly to the neuroendocrine differentiation of prostate cancer cells. This is a mis-leading statement. There are no data presented in the manuscript that show explicitly the necessity of MYCN mRNA regulation by ELAV3 to neuroendocrine differentiation of prostate cancer cells. More sophisticated approaches are needed to support this claim.

Figure 5. All of the problems that were noted with the experiments in Figure 4 are present in the data connecting ELAV3 to regulation of RICTOR mRNA. Accordingly, all of the same remedies need to be taken to offset these serious concerns.

Figure 6. The manuscript reports that pyrvinium pamoate (PP) inhibits growth of ELAV3 overexpressing cells in vitro and in vivo. This is interesting. But it is hardly an ELAV3-specific targeted agent. PP has been proposed to interfere with a slew of cancer relevant processes including Wnt signaling, electron transport and other mitochondrial functions, androgen receptor activity, hedgehog signaling, and the unfolded protein response. And ELAV1. It seems impossible, under these circumstances, to argue that ELAV3 is the target of PP. The RNA-Seq data in Figure 6G clearly shows many of these pathways are perturbed by PP treatment. More is needed to link PP action directly to ELAV3.

Figure 7. The type of EVs released from cells containing ELAV3 needs to be better and more specifically characterized.

Minor concerns:

Manuscript needs to be carefully revised for grammar.

The authors refer to ELAV3 as a "classic RNA binding protein". What does this mean?

Western blots are grainy and pixelated throughout.

**Authors' Response to the Reviewers' Comments on NCOMMS-23-15020-T**

We thank all the Reviewers for the time they spent in improving our work and for their
constructive comments. We have addressed all the Reviewers' concerns point-by-point. All
changes were highlighted in yellow in the revised manuscript.

**Reviewer #1 - Prostate cancer transcriptomics, RIP-seq, RNA-seq (Remarks to the**
**Author):**

*Major points:*

*Comment 1: ELAVL3 RIP-seq experiments were performed in PC-3 and DU145 cells with or*
*without ectopic expression of ELAVL3. Why did the authors perform such experiments in*
*H660 cells which express higher level of endogenous ELAVL3? At least some of the targets*
*of ELAVL3 be verified by RIP-qPCR in H660 cells.*

*Response:* We agree that data of endogenous RIP-qPCR in H660 cells are more convincing
than that in PC-3 and DU-145 cells. Unfortunately, despite testing all the five commercially
available ELAVL3 antibodies (ab210554 from Abcam, A-21272 from Invitrogen, PA5-96504
from Invitrogen, sc-515624 from Santacruz, and A6091 from Abclonal), none of them proved
suitable for immunoprecipitation. Therefore, we were compelled to perform RIP-seq and RIP-
qPCR with V5-tagged ELAVL3.

*Comment 2: By performing RIP-qPCR analysis, the authors showed that ELAVL3 binds to*
*MYCN mRNA. Can this observation be confirmed by the ELAVL3 RIP-seq data?*

*Response:* Thanks for bringing this to our attention. In our previous manuscript, we focused
on investigating the function of ELAVL3 and conducted RIP-seq to compare the ectopic
expression of ELAVL3 with vector control in both PC3 and DU145 cell lines. Our analysis
revealed enrichment of several oncogenic pathways in the overlapping immunoprecipitation
targets of both cell lines, underscoring the significant role of ELAVL3 in tumor development.
However, in these data, we did not observe specific peaks on MYCN and RICTOR mRNA—the
expected target of ELAVL3 in our study, not even VEGF—the reported target of ELAVL3¹, due
to insufficient sequencing depth. To better characterize the binding of ELAVL3 to target RNA,
we performed additional RIP-seq with strengthened sequencing depth in the revised
manuscript to compare the full length of ELAVL3 with RRM1-deletion ELAVL3, which lacks
the binding ability to target RNA. As anticipated, we observed specific peaks on MYCN and
RICTOR mRNA in the full-length ELAVL3 but not in the RRM1-deletion variant (revised Fig.
4e and 5c).

1 King, P. H. RNA-binding analyses of HuC and HuD with the VEGF and c-myc 3'-
 untranslated regions using a novel ELISA-based assay. *Nucleic Acids Research* 28, E20
 (2000).

*Comment 3: Explanation should be provided as to why different cell lines were used in*
 *different experiments such as RIP-seq, RIP-qPCR, ELAVL3 binding of MYCN mRNA and*
 *ELAVL3 regulation of MYCN mRNA stability.*

*Response: We thank the Reviewer for this comment. As mentioned above, there were no*
 *immunoprecipitation-grade antibodies available for ELAVL3, so we overexpressed V5-tagged*
 *ELAVL3 in at least two prostate cancer cell lines (PC3, LNCaP, and DU145) and performed*
 *RIP-seq as well as RIP-qPCR experiments with V5 antibody. For mRNA stability, we*
 *investigated the role of ELAVL3 by gain-of-function and loss-of-function experiments in PC3*
 *and LASCPC-01 cells, respectively. We apologize for not showing all the data in the original*
 *manuscript due to the limited space and added in the revised manuscript as following:*

- (1) RIP-seq experiments in PC3 and DU145 cells (Fig. 2a, b, Fig. 4e, and Fig. 5c);
- (2) RIP-qPCR experiments in LNCaP and PC3 cells (Fig. 4f, Fig. 5d, Extended Data Fig. 4e,
 Extended Data Fig. 5f, Fig. 6f, and Extended Data Fig. 6c);
- (3) RNA pulldown experiments in HEK293T, LNCaP, and PC3 cells (Fig. 4i and Fig. 5f);
- (4) mRNA decay experiments in PC3 and LASCPC-01 cells (Fig. 4k, Fig. 4m, Fig. 5i, Extended
 Data Fig. 4g, Extended Data Fig. 5h, i).

*Comment 4: The color differences for different RRM domain deletion mutations in Figure 4h*
 *are too subtle to be distinguished.*

*Response: We appreciate the Reviewer's suggestion and revised the colors of the graphs*
 *(revised Fig. 4d).*

*Comment 5: Is RICTOR mRNA a binding target of ELAVL3 which can be revealed by ELAVL3*
 *RIP-seq studies?*

*Response: As mentioned above, we observed the peaks of ELAVL3 on RICTOR mRNA in RIP-*
 *seq from PC3 cells overexpressing full length of ELAVL3, but not from RRM1-deletion one*
 *(revised Fig. 5c).*

*Comment 6: It is unclear that the anti-cancer effect of pyrvinium pamoate (PP) is mediated*
 *through ELAVL3.*

*Response: To assess the effect of pyrvinium pamoate (PP) on ELAVL3's function, we first*
 *compared the cell viability in LASCPC-01 cells with increasing concentration of PP and*
 *showed that shRNA-ELAVL3 reduced cell sensitivity to PP compared with shRNA control (Fig.*
 *6d). To further examine the effect of PP on ELAVL3's activity, we performed luciferase*
 *reporter assay driven by full-length MYCN 3'-UTR and showed that PP treatment reversed*
 *the elevation of luciferase signal induced by ELAVL3 (Fig. 6e). In RIP-qPCR assay, PP*
 *treatment also reversed the enrichment of MYCN mRNA by ELAVL3 (Fig. 6f). Most*
 *importantly, we performed RNA Electrophoretic Mobility-Shift Assay (RNA-EMSA) with*
 *recombinant ELAVL3 protein and biotinylated MYCN or RICTOR 3'-UTR RNA probes. Our*
 *results revealed a clear shift band due to ELAVL3 binding, whereas PP treatment led to a*
 *dramatical decrease in ELAVL3 binding (revised Fig. 6g, shown below). All above results*
 *strongly suggest that the anti-cancer effect of PP is mediated through ELAVL3.*

*Minor points:*

*Comment 7: English and grammar need to be polished and/or edited.*

*Response: We carefully improved the language in our revised manuscript.*

**Reviewer #2 - Prostate cancer differentiation, vesicles (Remarks to the Author):**

**MAJOR POINTS**

*Comment 1: The immunohistochemical analysis is performed with an antibody that needs*
*to be validated by competition with the antigen/antigen peptide detected*

*Response: We validated the specificity of the ELAVL3 antibody by immunohistochemical*
*staining using a commercial recombinant ELAVL3 protein (CSB-YP618895HU, CUSABIO)*
*(revised Extended Data Fig. 2h).*

*Comment 2: Fig 1f indicates that the H660 cell line expresses high levels of ELAVL3; some*
*of the other cell lines express a lower MW band. How do the Authors explain this discrepancy?*

*Response: We did observe two bands of ELAVL3 in different cell lines with several antibodies:*
*one is around 40kD and the other is between 35 kD and 40 kD (revised Fig. 1f, shown below).*
*This observation has been previously described with Western blot in tissues of spin cords*
*from healthy donor and amyotrophic lateral sclerosis patients^{1,2}, showing that there are two*
*different bands of ELAVL3 (PMID:34618203, PMID:30643292). The antibody datasheets (for*
*PA5-96504 from Invitrogen, sc-515624 from Santacruz, and A6091 from Abclonal, shown*
*below) also provide evidences supporting this observation. In their 2020 *Nat Neurosci**
*manuscript, Klim reported the presence of a cryptic exon in ELAVL3, which may explain these*
*results². Therefore, we hypothesize that ELAVL3 may be post-transcriptionally regulated in*
*H660 cells, which required additional experiments but are beyond the scope of this*
*manuscript.*

PMID:34618203

<https://www.scbt.com/p/huc-antibody-d-6/>

1 Diaz-Garcia, S. et al. Nuclear depletion of RNA-binding protein ELAVL3 (HuC) in sporadic
and familial amyotrophic lateral sclerosis. *Acta Neuropathol* 142, 985-1001,
doi:10.1007/s00401-021-02374-4 (2021).

2 Klim, J. R. et al. ALS-implicated protein TDP-43 sustains levels of STMN2, a mediator of
motor neuron growth and repair. *Nat Neurosci* 22, 167-179, doi:10.1038/s41593-018-0300-
4 (2019).

*Comment 3: Fig 2 shows only 1shRNA tested in vivo*

Response: We add experiments for another shRNA sequences in the revised manuscript. The
results are consistent: ELAVL3 ablation inhibits tumor growth in NEPC mice model (revised
Fig. 2h, i, shown below).

*Comment 4: The weak specificity of the drug used, PP, is a concern*

Response: To assess the drug specificity of pyrvinium pamoate (PP) in neuroendocrine
 prostate cancer, we performed the cell viability in LASCPC-01 cells with increasing
 concentration of PP and showed that shRNA-ELAVL3 reduced cell sensitivity to PP compared
 with shRNA control (revised Fig. 6d). To further examine the effect of PP on ELAVL3's activity,
 we performed a luciferase reporter assay driven by full-length MYCN 3'-UTR and showed that
 PP treatment reversed the elevation of the luciferase signal induced by ELAVL3 (revised Fig.
 6e). In RIP-qPCR assay, PP treatment also reversed the enrichment of MYCN mRNA by
 ELAVL3 (revised Fig. 6f). Most importantly, in the revised manuscript, we performed RNA
 Electrophoretic Mobility-Shift Assay (RNA-EMSA) with recombinant ELAVL3 protein and
 biotinylated MYCN or RICTOR 3'-UTR RNA probes and showed clear shift bands due to the
 binding of ELAVL3, whereas PP treatment led to a dramatical decrease in ELAVL3 binding
 (revised Fig. 6g, shown below). These results confirmed that the anti-cancer effect of PP is
 mediated through its interaction with ELAVL3.

*Comment 5: The Extended 5b image is not optimal and the data are not quantified*
 *Response: We repeated the experiments and improved the image quality of the Extended*
 *5b. Our conclusion remains the same that PP has little effects on the subcellular localization*
 *of ELAVL3 (revised Extended Data Fig. 6b, shown below).*

 *Comment 6: The data in the Extended 6d need to be supported by an immunoblot*
 *Response: We have now added immunoblot data in the revised manuscript, showing that*
 *full-length but not RRM1-deletion of ELAVL3 upregulates MYCN, RICTOR, and NE markers*
 *(revised Extended Data Fig. 7e, shown below).*

 *Comment 7: Fig 6d: protease treatment is needed to confirm that the protein is inside the*
 *extracellular vesicles*
 *Response: As suggested, we treated the extracellular vesicles with proteinase K and*
 *confirmed that ELAVL3 are encapsulated inside the extracellular vesicles. Our results remain*
 *the same: the neuroendocrine markers in target cells were upregulated by EVs from PC3*
 *cells expressing full-length ELAVL3 but not from RRM1-deletion mutation (revised Extended*
 *Data Fig. 7e, shown below).*

 *Comment 8: Fig 6: Further isolation of extracellular vesicles by density gradient should be*
 *used to confirm association of ELAVL3 with extracellular vesicles*

Response: As suggested, we performed density gradient isolation on EV pellets and
 confirmed the association of ELAVL3 with extracellular vesicles in revised Extended Data Fig.
 7d, shown below. We have added the following to the manuscript methods: EV pellets were
 re-suspended in 1.5 mL of suspension buffer (0.25M sucrose, 10mM Tris pH 8.0, 1mM EDTA
 pH 7.4) and mixed in a 1:1 ratio with a 60% stock solution of iodixanol/Optiprep (D1556,
 Sigma Aldrich). Stepwise layering of different iodixanol concentrations was then performed
 as follow: 1.4 mL of 40% iodixanol, 1.3 mL of 20% iodixanol, and 1.2 mL of 10% iodixanol
 were gently layered on top of the EV suspension. Subsequently, the tubes were
 ultracentrifuged at 120,000 × g overnight at 4°C. Then, ten fractions of 700 μL each were
 collected from the top of the tube individually, washed with PBS, and subjected to
 ultracentrifugation at 120,000 × g for 2 hours at 4°C. Finally, the pellets were re-suspended
 in 40 μL of lysis buffer for Western Blot.

*Comment 9: TKO mice are not characterized*

Response: As recommended, we described the construction and genotyping of TKO mice in
 Methods section and revised Extended Data Fig. 8a, shown below.

*Comment 10: The Extended 6e data are not described; is ELAVL3 absent in those samples?*

Response: We apologize for this mistake and revised the manuscript. "However, we did not
 observe this increase of ELAVL3 on the protein level, which may due to the trace amount in
 patient sera (Extended Data Fig. 7f)".

*Reviewer #3 - MYC, drug screens (Remarks to the Author):*

*Major concerns:*

*Comment 1: Figure 1F. The authors claim high basal level expression of ELAVL3 in LASCPC-*
 *01 and H660 cells at the protein level. But there is no apparent expression of ELAVL3 in any*
 *of the lines except H660. As LASCPC-01 cells are NEPC-derived, the absence of ELAVL3*
 *expression in this setting challenges the generality of the results presented in the manuscript.*
 *This issue is further problematic due to the the fact that H660 cells express high levels of*
 *ELAVL3, but have no detectable MYCN protein expression.*

*Response: We thank the Reviewer for this comment and apologize for the inconsistencies.*
 *We have regenerated the Western Blot in Fig. 1f with biological triplicates (n=3) to measure*
 *the protein levels for ELAVL3 in a panel of prostate cancer cell lines. We did observe high*
 *level of ELAVL3 in both LASCPC-01 and H660 cells, consistent with results from qPCR*
 *showing that H660 has a higher RNA level of ELAVL3 than LASCPC-01 (shown below).*
 *We also regenerated the Western Blot under short or long exposure time, showing the*
 *expression of MYCN in LASCPC-01 and H660 cells. Since LASCPC-01 was established by*
 *ectopic expression of MYCN and activated AKT1, it is not surprising that both the RNA and*
 *protein levels of MYCN were extremely high. While MYCN protein levels in H660 were lower*
 *than in LASCPC-01, it remained comparatively higher than other prostate cancer cells.*

1 Lee, J. K. et al. N-Myc Drives Neuroendocrine Prostate Cancer Initiated from Human
 Prostate Epithelial Cells. *Cancer Cell* 29, 536-547, doi:10.1016/j.ccell.2016.03.001 (2016).

*Comment 2: Figure 2D. shRNA knockdown of ELAVL3 in H660 cells needs to be done with a*
 *second shRNA (as in LASCPC-01 cells). A western blot needs to also be performed to show*
 *knockdown at the protein level.*

*Response: We knocked down ELAVL3 in H660 with a second shRNA sequence and showed*
 *both the RNA and protein levels in revised Fig. 2d and Extended Data Fig. 2c, shown below.*

*Comment 3: Figure 2E. The western blot on LASCPC-01 cells in this part of the figure appears*

at odds with that shown in Fig. 1F. In the latter, we see high levels of MYCN and no
expression of ELAV3. Here, we see expression of ELAV3 but MYCN appears barely above
background. What is the cause of this discrepancy?

Response: We apologize for the confusion. The reason for the discrepancy between Fig. 1F
and Fig. 2e was due to difference in exposure time. Given the extremely high expression of
MYCN in LASCPC-01, we reduced the exposure time in Fig. 2e to demonstrated the
downregulation of MYCN by ELAVL3 ablation in previous manuscript. Here we regenerated
the Western Blot and our results remain the same—that ELAVL3 is increased in LASCPC-01
and H660 and that knockdown of ELAVL3 can reduce the expression of MYCN.

*Comment 4: Figure 2J and L. The data connecting ELAV3 expression to enzalutamide are*
*weak. The effects of ELAV3 knockdown on LAPR viability and response to enzalutamide are*
*subtle, and the statement that ELAV3 "facilitates the acquisition of enzalutamide resistance"*
*is made without sufficient justification. The conclusions need to be toned down.*

Response: We thank the Reviewer for this suggestion and have revised the results section:
Collectively, these results indicate that ELAVL3 sustains the proliferation of neuroendocrine
prostate cancer cells and modestly reduces the cell response to enzalutamide.

*Comment 5: Figure 3. One of the key conclusions in the manuscript is that ELAV3 is a direct*
*transcriptional target of MYCN. Indeed, this is the bedrock of the story. Yet the data*
*supporting this conclusion are weak, and based largely on (i) transient reporter assays that*
*do not necessarily reflect faithfully reflect events in endogenous genes, (ii) long term*
*knockdown experiments that conflate primary, secondary, and tertiary effects, and (iii)*
*indirect measures of transcriptional output. The flimsy nature of these data undermines the*
*major conclusions of the work. To support this conclusion, candidate E-box elements need*
*to be mutated in the endogenous ELAV3 promoter, acute perturbations of MYCN need to be*
*used, and some measure of transcription (not transcripts, not reporter proteins) needs to be*
*performed. The same general criticism applies to the indirect/artificial assays used to*
*probe regulation of MYCN mRNA in Figure 4.*

Response: We thank the Reviewer for this constructive criticism, which was extremely
valuable to further strengthen our work. As suggested, we applied CRISPR/Cas9-mediated
system to target the endogenous MYCN binding motif at the promoter region of ELAVL3. Two
sgRNAs were designed to target the MYCN binding sites in the ELAVL3 promoter region and

the non-targeting sgRNA were used as control (revised Extended Data Fig. 3b, shown below).
 Next, we used 4-thiouridine (4sU) to label the nascent RNA, which was then isolated by
 biotin pull-down and quantified by QPCR. Our results showed that ectopic expression of
 MYCN increased the nascent ELAVL3 transcription, which was rescued by CRISPR/Cas9-
 mediated deletion of MYCN binding PAM sites (revised Fig. 3g, shown below). These results
 strongly suggested that ELAVL3 is a direct transcriptional target of MYCN.
 However, this strategy could not be used for MYCN mRNA 3'UTR. As showed in Extended
 Data Fig. 4g, ELAVL3 can prolong the half-life of MYCN mRNA without 3'-UTR. Thus, we
 conclude that ELAVL3 may also bind to MYCN coding sequence. Based on this, even we
 knockout or mutate the endogenous 3'-UTR of MYCN, ELAVL3 still can binds to and regulate
 MYCN.

*Comment 6: Figure 4A. Western blots need to be included to show knock-down of MYCN at*
 *the protein level. We also need to see how this impacts ELAV3 protein expression.*

*Response: As suggested, we performed the Western Blot to show the knockdown of MYCN*
 *and the reduction of ELAVL3 protein (revised Fig. 4a, shown below).*

*Comment 7: Figure 4F. The authors conclude that ELAV3 directly binds to the MYCN mRNA.*
 *This, again, is a foundation of the story in the manuscript. Yet there are no direct ELAV3-*
 *mRNA binding assays here; just a pulldown from crude lysates. The authors need to show a*
 *direct interaction between recombinant ELAV3 and purified mRNA to support this important*

claim.

Response: We thank the Reviewer for raising this issue. We performed RNA-EMSA assay with

recombinant ELAVL3 protein to confirm the interaction of ELAVL3 and MYCN 3'-UTR. Our

results clearly showed that the purified GST-ELAVL3 can bind to the biotin-labeled MYCN-

specific RNA probe, but not to the mutated probe (revised Fig. 4j, shown below). This in

vitro assay provides strong evidence that ELAVL3 directly interacts with MYCN 3'-UTR mRNA.

Comment 8: Figure 4G. An actinomycin D assay is used to infer that ELAV3 regulates the

stability of MYCN mRNA. These assays are prone to artifacts associated with the use of a

potent transcriptional poison. The effect on MYCN mRNA stability needs to be shown with a

second, more focused (non-poison) approach such as pulse-labeling or a specific promoter

shutoff.

Response: We appreciate the Reviewer for this valuable suggestion. As suggested, we

performed a pulse-labeling assay in PC3/MYCN-3'-UTR cells with 4sU and chased for 8 hours.

The RNA was collected and processed to detect the degradation rate. Our results showed

that the half-life of 4sU-labelled MYCN mRNA was extended in cells expressing ELAVL3,

compared with vector control (revised Fig. 4l, m, shown below), indicating that ELAVL3

stabilize the mRNA of MYCN.

Comment 9: This second of the manuscript concludes with the statement that ties regulation

of MYCN mRNA stability directly to the neuroendocrine differentiation of prostate cancer cells.

This is a mis-leading statement. There are no data presented in the manuscript that show

explicitly the necessity of MYCN mRNA regulation by ELAV3 to neuroendocrine differentiation

of prostate cancer cells. More sophisticate approaches are needed to support this claim.

Response: We appreciate the Reviewer for this constructive comment. Previous studies have

highlighted the critical role of MYCN in driving neuroendocrine differentiation in prostate

cancer¹⁻³. On the basis of these, our study reveals that ELAVL3 is essential for initiating and
maintaining of NEPC. We have demonstrated the significance of ELAVL3 and MYCN in this
progression. Importantly, in the revised manuscript, we have provided evidence that
overexpression of ELAVL3 induces neuroendocrine differentiation, as measured by NE
markers in prostate cancer cells (revised Fig. 4a). Notably MYCN deficiency reverses this
elevation, underscoring the requirement for MYCN in ELAVL3-induced phenotype.
Furthermore, we generated ELAVL3 mutated constructs ELAVL3-RRM1Δ and -RRM2Δ, both
of which lack the ability to bind downstream RNA (revised Fig. 4d). Our results showed that
these mutant constructs fail to induce neuroendocrine differentiation (revised Fig. 4c).
Altogether, these findings strongly suggest that the regulation of MYCN by ELAVL3 is required
for ELAVL3-induced neuroendocrine differentiation.

1 Dardenne, E. et al. N-Myc Induces an EZH2-Mediated Transcriptional Program Driving
Neuroendocrine Prostate Cancer. *Cancer Cell* 30, 563-577, doi:10.1016/j.ccell.2016.09.005
(2016).

2 Lee, J. K. et al. N-Myc Drives Neuroendocrine Prostate Cancer Initiated from Human
Prostate Epithelial Cells. *Cancer Cell* 29, 536-547, doi:10.1016/j.ccell.2016.03.001 (2016).

3 Yin, Y. et al. N-Myc promotes therapeutic resistance development of neuroendocrine
prostate cancer by differentially regulating miR-421/ATM pathway. *Mol Cancer* 18, 11,
doi:10.1186/s12943-019-0941-2 (2019).

*Comment 10: Figure 5. All of the problems that were noted with the experiments in Figure*
*4 are present in the data connecting ELAV3 to regulation of RICTOR mRNA. Accordingly, all*
*of the same remedies need to be taken to offset these serious concerns.*

Response: As suggested, we performed an RNA-EMSA assay with recombinant ELAVL3
protein and RICTOR-specific RNA probes to confirm the interaction between ELAVL3 and
RICTOR 3'-UTR. Our results showed that the purified GST-ELAVL3 can bind to the biotin-
labeled RICTOR-specific RNA probe, but not the mutated probe, indicating a direct interaction
between ELAVL3 and RICTOR mRNA (revised Fig. 5g, shown below). Moreover, we performed
a pulse-labeling assay in the PC3 cells with 4sU and chased for 8 hours to detect the
degradation of the mRNA (revised Extended Data Fig. 5i, shown below). The results showed
that overexpression of ELAVL3 extend the half-life of 4sU-labeled RICTOR mRNA, while
knockdown of ELAVL3 did the opposite, indicating that ELAVL3 stabilize the mRNA of RICTOR.
Additionally, we either knocked down RICTOR or used the AKT pathway inhibitor MK-2206
on PC3/ELAVL3 cells (revised Fig. 5j, k and revised Extended Data Fig. 5j, k, shown below).
The results showed that both genetically and pharmacologically inhibiting RICTOR could
decrease neuroendocrine marker which is induced by ELAVL3 overexpression.

GST	+	-	-	-	-	-
GST-ELAVL3 (μM)	-	-	0.5	1	2	2
F3 WT probe	+	+	+	+	+	-
Nonlabelled F3 probe (20 \times)	-	-	-	-	-	+
F3 MUT probe	-	-	-	-	-	+

*Comment 11: Figure 6. The manuscript reports that pyrvinium pamoate (PP) inhibits growth*
 *of ELAV3 overexpressing cells in vitro and in vivo. This is interesting. But it is hardly an*
 *ELAV3-specific targeted agent. PP has been proposed to interfere with a slew of cancer*
 *relevant processes including Wnt signaling, electron transport and other mitochondrial*
 *functions, androgen receptor activity, hedgehog signaling, and the unfolded protein*
 *response. And ELAV1. It seems impossible, under these circumstances, to argue that ELAV3*
 *is the target of PP. The RNA-Seq data in Figure 6G clearly shows many of these pathways*
 *are perturbed by PP treatment. More is needed to link PP action directly to ELAV3.*

Response: We thank the Reviewer for this critical comment. To interrogate the impact of
 PP to ELAVL3, we performed an RNA Electrophoretic Mobility-Shift Assay (RNA-EMSA) with
 recombinant ELAVL3 protein and biotinylated MYCN or RICTOR 3'-UTR RNA probes and
 showed a clear shift band due to the binding of ELAVL3, whereas PP treatment led to a
 dramatical decrease in ELAVL3 binding (revised Fig. 6g, shown below). These findings
 strongly suggest the direct disruption of PP on ELAVL3 RNA-binding function.

*Comment 12: Figure 7. The type of EVs released from cells containing ELAV3 needs to be*
 *better and more specifically characterized.*

Response: As suggested, we performed density gradient isolation to characterize the EV
 pellets and demonstrated the association of ELAVL3 with extracellular vesicles in revised
 Extended Data Fig. 6d, shown below. We have added the following to the manuscript
 methods: EV pellets were re-suspended in 1.5 mL of suspension buffer (0.25M sucrose,
 10mM Tris pH 8.0, 1mM EDTA pH 7.4) and mixed in a 1:1 ratio with a 60% stock solution of
 iodixanol/Optiprep (D1556, Sigma Aldrich). Stepwise layering of different iodixanol
 concentrations was then performed as follow: 1.4 mL of 40% iodixanol, 1.3 mL of 20%
 iodixanol, and 1.2 mL of 10% iodixanol were gently layered on top of the EV suspension.
 Subsequently, the tubes were ultracentrifuged at 120,000 × g overnight at 4°C. Then, ten
 fractions of 700 μL each were collected from the top of the tube individually, washed with
 PBS, and subjected to ultracentrifugation at 120,000 × g for 2 hours at 4°C. Finally, the
 pellets were re-suspended in 40 μL of lysis buffer for Western Blot.

*Minor concerns:*

*Comment 13: Manuscript needs to be carefully revised for grammar.*

Response: We carefully improved the language in our revised manuscript.

*Comment 14: The authors refer to ELAV3 as a "classic RNA binding protein". What does this*
 *mean?*

Response: Sorry for the confusion. In the original manuscript, we intended to emphasize the
RNA-binding function of ELAVL3. To minimize the confusion, we deleted "classic" from the
sentence, which now reads: "To investigate the role of ELAVL3 in neuroendocrine prostate
cancer, we performed ribonucleoprotein immunoprecipitation sequencing (RIP-seq) analyses
in PC3 and DU145 cells overexpressing ELAVL3."

*Comment 15: Western blots are grainy and pixelated throughout.*

Response: We have provided revised Western blots images with better qualities (revised Fig.
1f, 2c, 2e, 4a, 4i, 5f, 6c, 8n, Extended Data Fig. 2b, 2c, 2e, 4d, 5k, 7e).

Reviewers' Comments:

Reviewer #1:

Remarks to the Author:

The authors have adequately addressed all my concerns and the revised manuscript is now suitable for publication in Nature Communications.

Reviewer #2:

Remarks to the Author:

All concerns are fully addressed in the revised manuscript.

Reviewer #3:

Remarks to the Author:

I think in general the manuscript has been dramatically improved and the story strengthened in revision. Specific comments:

Comment 1. Figure 1F. The new blots satisfy this concern.

Comment 2. Figure 2D. Addition of a second shRNA is excellent. Results consistent. Addition of western informative.

Comment 3. Figure 2E. New exposures and explanation both satisfactory.

Comment 4. Figure 2J and L. Conclusion satisfactorily toned down.

Comment 5. Figure 3. Excellent addition of E-box mutant and 4sU labeling. Very significant improvement.

Comment 6. Figure 4A. Western blots now included.

Comment 7 Figure 4F. Addition of the EMSA is an important and convincing addition.

Comment 8. Figure 4G. Great addition. But is the difference in MYCN mRNA stability significant? Statistical tests are given for other data sets in this figure, but not 4M.

Comment 9. Satisfactory response.

Comment 10. Figure 5. The new data are excellent and compelling. Statistics are needed on the mRNA turnover data.

Comment 11. Figure 6. EMSA is excellent and compelling.

Comment Figure 7. This is a very superficial characterization. I recommend removing, as it's too preliminary.

Grammar is significantly improved.

Western blots are improved but some are still quite grainy. May just be in the version I reviewed.

**Authors' Response to the Reviewers' Comments on NCOMMS-23-15020A**

We express our sincere gratitude to all the reviewers for their invaluable time and
constructive feedback. Each of their concerns has been meticulously addressed in our
updated manuscript.

**Reviewer #1 (Remarks to the Author):**

*The authors have adequately addressed all my concerns and the revised manuscript is*
*now suitable for publication in Nature Communications.*

Response: We thank the reviewer for his/her time and efforts in evaluating the revised
manuscript.

**Reviewer #2 (Remarks to the Author):**

*All concerns are fully addressed in the revised manuscript.*

Response: We thank the reviewer for his/her time and efforts in evaluating the revised
manuscript.

**Reviewer #3 (Remarks to the Author):**

*I think in general the manuscript has been dramatically improved and the story*
*strengthened in revision. Specific comments:*

Response: We sincerely appreciate the reviewer for dedicating time and effort to review the
updated manuscript and for providing a positive evaluation. We have addressed the issues
highlighted by the reviewer and have supplemented the manuscript with additional
supporting data.

*Comment 8. Figure 4G. Great addition. But is the difference in MYCN mRNA stability*
*significant? Statistical tests are given for other data sets in this figure, but not 4M.*

Response: We appreciate the suggestion and have now included biological replicates to
provide statistical significance both in revised Figure 4k and 4m, shown below.

*Comment 10. Figure 5. The new data are excellent and compelling. Statistics are needed*
*on the mRNA turnover data.*

Response: Based on this suggestion, we have added statistical significance in revised Figure
5i.

*Comment Figure 7. This is a very superficial characterization. I recommend removing, as*
 *it's too preliminary.*

*Response:* We thank the reviewer for this feedback. We agree with the reviewer that further
 investigation and a broader perspective are warranted concerning the results presented in
 Figure 7. Nevertheless, we maintain that our current findings offer a foundational and
 indispensable insight. Specifically, they demonstrate that ELAVL3 is released in EVs and
 induce neuroendocrine differentiation of recipient adenocarcinoma cells. These discoveries
 are not only consistent with previous studies on EV-mediated intercellular communication
 between neuroendocrine prostate cancer cells and adenocarcinoma cells, but also suggest a
 promising avenue wherein ELAVL3 may serve as a potential serum biomarker for the
 diagnosis of neuroendocrine tumors.

*Western blots are improved but some are still quite grainy. May just be in the version I*
 *reviewed.*

*Response:* We thank the reviewer for pointing out the concerns regarding the clarity of some
 Western blots in our manuscript. We would submit a high-resolution PPT file to ensure the
 clarity and quality of the images. It's possible that there might have been some resolution
 loss during the conversion into PDF file for reviewer.